Corrected: Author correction

# The temporal dynamics and infectiousness of subpatent *Plasmodium falciparum* infections in relation to parasite density

Hannah C. Slater et al.[#]

Malaria infections occurring below the limit of detection of standard diagnostics are common in all endemic settings. However, key questions remain surrounding their contribution to sustaining transmission and whether they need to be detected and targeted to achieve malaria elimination. In this study we analyse a range of malaria datasets to quantify the density, detectability, course of infection and infectiousness of subpatent infections. Asymptomatically infected individuals have lower parasite densities on average in low transmission settings compared to individuals in higher transmission settings. In cohort studies, subpatent infections are found to be predictive of future periods of patent infection and in membrane feeding studies, individuals infected with subpatent asexual parasite densities are found to be approximately a third as infectious to mosquitoes as individuals with patent (asexual parasite) infection. These results indicate that subpatent infections contribute to the infectious reservoir, may be long lasting, and require more sensitive diagnostics to detect them in lower transmission settings.

The occurrence of malaria infections below the parasite density threshold detectable by conventional field diagnostics (microscopy and rapid diagnostic tests (RDT)) is well established[1]. These infections are considerably less likely to cause symptoms[2]; however, in feeding assays they have been shown to be infectious and are likely to contribute to the infectious reservoir and onwards transmission[3–5]. Furthermore, parasite densities in infected individuals fluctuate over time meaning that these low-density infections may have future phases of higher density, infectiousness and potentially morbidity[6–8]. As the drive towards malaria elimination intensifies, the question arises as to the added benefit of identifying and treating low-density infections in order to interrupt transmission. The answer to this lies in better understanding the prevalence, detectability, temporal dynamics and infectivity of these low-density infections in a range of transmission settings.

Here, for clarity, we outline the terminology used throughout this article: a submicroscopic infection is defined as an infection which is detectable by molecular methods, but not by microscopy. A subpatent infection is an infection which is detectable by molecular methods, but not by the field diagnostic being used in the study, whether this is microscopy or RDT. A low-density infection is as described, with no reference to the diagnostic being used to try and detect it. A submicroscopically or subpatently infected individual is someone who is detectable by molecular methods but is undetectable by microscopy or microscopy/RDT, respectively. All these definitions refer specifically to infections with asexual parasites. Most of the studies analysed here use a form of PCR as their molecular method, therefore, for brevity, we use the term PCR synonymously with 'molecular methods'.

A large number of surveys confirm the existence of subpatently infected individuals in all transmission settings. However, the proportion of all PCR-detected malaria-infected individuals that are detected by microscopy/RDT is significantly lower in lower transmission settings. This indicates a lower average parasite density in these settings, contrary to what might be expected given the lower antiparasite immunity to malaria in these populations[1,9,10]. Until recently, most PCR methods detected presence or absence of infection but were not quantitative, and therefore the distribution of parasite densities in the subpatent range was unknown. Wider application of quantitative molecular methods now offers the opportunity to study how parasite density distributions in infected populations may shift as transmission is reduced. Combining parasite density distribution data with infectivity data allows us to estimate how much of the infectious reservoir will be detectable with more sensitive diagnostics.

To date, most reviews of low density, subpatent infections focus on cross-sectional prevalence surveys.[1] While informative, these offer only a snapshot in time of an otherwise a dynamic situation. When initially infected, a person has a period of subpatent parasitaemia after blood-stage parasites emerge from the liver, before they multiply to microscopically detectable densities. This takes longer in semi-immune adults (average ~3 weeks, sometimes >63 days) compared with young children (average 1 week)[11], and indeed some infections may never reach microscopically detectable levels. Untreated *P. falciparum* infections persist on average for 6 months[8,12,13], but can last anywhere between a few weeks and several years[6], and parasite density fluctuates over the course of an infection. Artificially induced malariatherapy infections showed clear declines in average parasite density and detectability over time even when no treatment was given[1,14,15]. However in these infections, low-density periods also occurred early in infection, with more than 60% of untreated patients experiencing a subpatent period lasting for 1–34 days directly after the initial peak in parasitaemia[16]. In naturally acquired infections, changes in detectability in

relation to duration of infection are extremely difficult to measure due to superinfection, where individuals may be repeatedly infected over time with different parasite strains[17]. This makes it difficult to measure the relative densities or durations of different parasite clones without longitudinal genetic studies. Therefore a better understanding of these dynamics in endemic, semi-immune populations is key to elucidating whether there is a need to identify and treat low-density infections. If low-density infections are mainly short lived, destined to clear quickly without treatment, and not very infectious, perhaps their contribution to the infectious reservoir is small and clearing them is not essential to achieve elimination. Alternatively, if low-density phases occur frequently throughout an infection, treatment of low-density infections may prevent long periods of these individuals being infected and infectious. Additionally, low-density infections have been identified as a potential marker of micro heterogeneity of transmission (i.e., hotspots); therefore, treating and prophylactically protecting these individuals could prevent future morbidity and onwards transmission as these individuals are more likely to develop future symptomatic infections[18].

In this article we analyse a series of datasets, harnessing the increasing quantities of molecular data generated in recent years, to investigate the density, temporal dynamics and infectiousness of low-density *P. falciparum* infections. Firstly, we examine the parasite density distributions of asymptomatically infected populations in a range of transmission settings measured by quantitative molecular methods, with implications for the required sensitivity of new diagnostics. Next, longitudinal studies are analysed to elucidate the temporal dynamics of parasite densities in endemic populations, with particular reference to the risk of subpatently infected individuals developing future higher density infections. In the final section, we use studies where mosquito were fed on asymptomatically infected individuals to estimate the relative infectivity of individuals with subpatent parasitaemia and their contribution to the infectious reservoir.

## Results

**Density and detectability of *P. falciparum* infections**. A literature search was conducted to identify cross-sectional data on the parasite densities of individuals with *Plasmodium falciparum* infections in endemic areas measured using a quantitative molecular method. Fifteen articles were identified, consisting of data from 22 locations in a wide range of transmission intensities (prevalence by molecular method ranging from 0.4% to 90.6%)[3,4,12,13,19–30]. Details of the literature search criteria and more information on the diagnostic methods and study settings are in Methods and Supplementary Table 1.

The median parasite density in identified dataset was between 1 and 1336 parasites per µl (Fig. 1a). The range of parasite densities (parasites per µl) in infected individuals spanned at least two orders of magnitude, and in several cases, more than six orders of magnitude (Fig. 1a). Individuals in each dataset were split into two groups: those with infections that were detectable by PCR and microscopy or RDT, and those with infections that were detectable by PCR only (Fig. 1b). Median parasite density is lower in the subpatently infected individuals in all datasets compared with the microscopy/RDT-positive group. However, the interquartile range of the two groups overlaps in 3/18 of the datasets and the total range overlaps in all datasets. The limit of detection for microscopy in routine diagnostic settings is thought to be around 100 parasites per µl[31] and around 10 parasites per µl by expert microscopists, yet 12/18 datasets have subpatent infected individuals with parasite densities >100 parasites per µl and all have subpatent-infected individuals with parasite densities >10 parasites per µl. Infected individuals that are patent

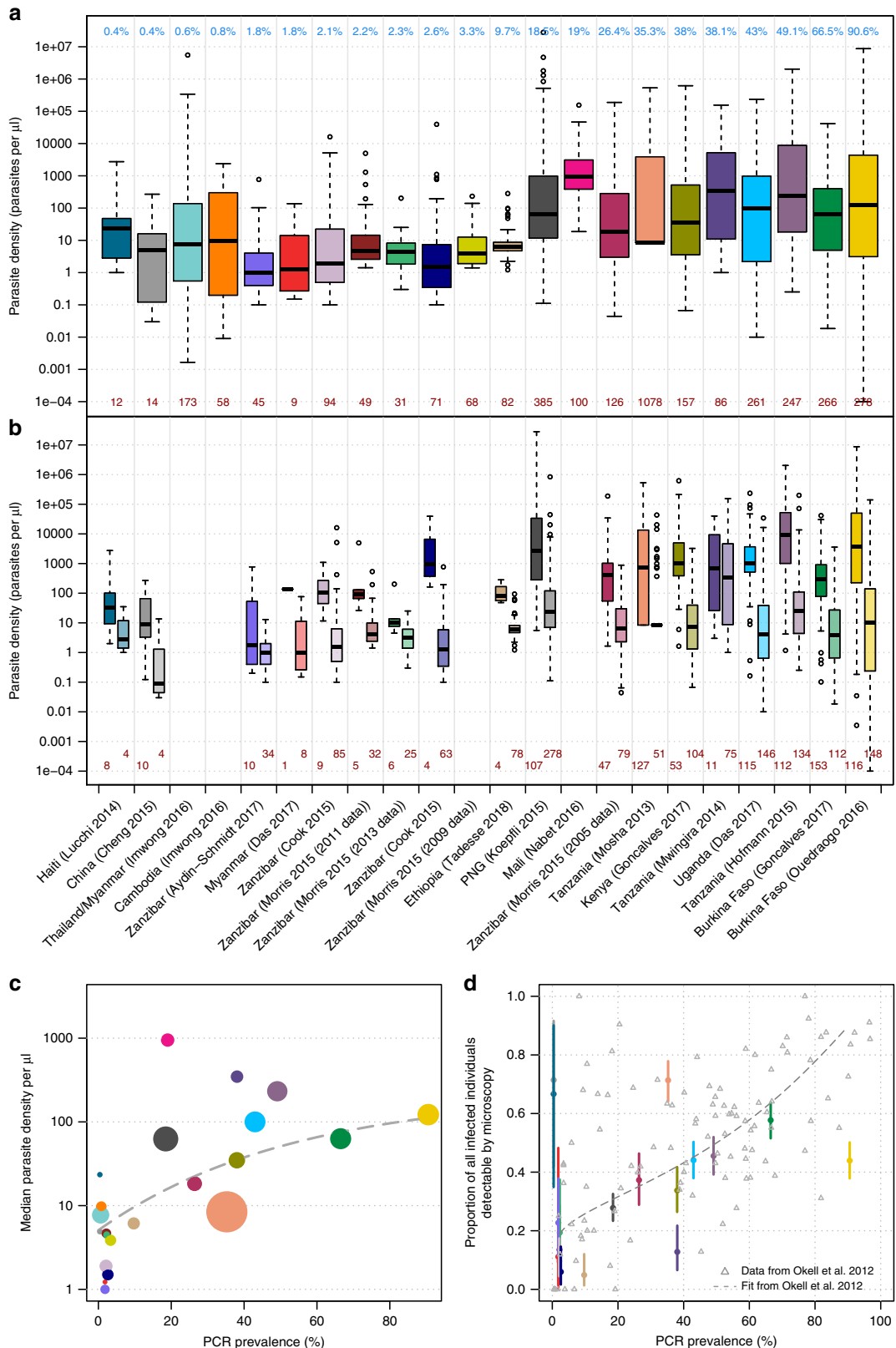

but have very low-parasite densities by PCR typically have asexual parasite densities by microscopy in the traditional range of this method (> 50 parasites per μl), indicating a disagreement in density measures between the two tests.

Figure 1c shows that median parasite density (from each of the studies in Fig. 1a) increases in higher transmission settings; a trendline fitted to the data indicates that the median parasite density increases from below five parasites per μl in the datasets with the lowest prevalence to 100 parasites per μl in the highest transmission datasets. As has been shown previously[1], the proportion of infected individuals that are detectable by microscopy/RDT is lowest in low transmission settings (Fig. 1d).

**Fig. 1** Parasite densities and detectability of asymptomatically infected individuals. **a** Boxplot of the parasite densities (parasites per μl) of all infected individuals by quantitative PCR or nucleic method. The blue numbers along the top indicate the PCR prevalence in each setting and the studies are ordered from left to right by prevalence. The dark red numbers along the bottom indicate the number of PCR-positive individuals in each study. The location and first author of each study is presented at the bottom of panel **b**. **b** The individuals in each study are separated in to two groups—those detectable by PCR and RDT/microscopy (darker shade on the left in each column) and those only detectable by PCR (lighter shade on the right). The dark red numbers along the bottom indicate the number of infected individuals in each group. For both boxplots, the centre line indicates the median, the upper and lower bounds of the box show the 25% and 75% percentiles, and the whiskers show the minimum and maximum values of each dataset. **c** The median parasite density in each study (also shown as the centre lines in the boxplots in panel **a** plotted against PCR prevalence. The fitted line is of the form: mean parasite density per μl = $a - b*exp(m*PCR$ prevalence) where $a = 2.342$, $b = 1.637$, $m = -1.896$. The size of each circle is proportional to the number of PCR-positive samples in each study (shown at the bottom of panel **a**). **d** The proportion of PCR-positive individuals that are also detected by microscopy/RDT and 95% binomial confidence intervals. These are compared to a previously published relationship[1], shown by the grey dashed line. The colours of the points correspond to the studies in panels 1a, b. Source data are provided as a Source Data file

We did not standardise the age distribution of data across the studies (as several studies did not include age data), but results from an age-standardised analysis were consistent with those in Fig. 1 (Supplementary Note 1 and Supplementary Figure 3). There was no difference in the parasite densities of submicroscopic compared with sub-RDT infections from an analysis looking at the two studies which used both diagnostics and had sufficient sample sizes (Supplementary Figure 2).

A Bayesian logistic regression model with a study-level random effect was used to estimate the relationship between parasite density by quantitative PCR and the probability of detection by microscopy. We used a normal prior distribution with a mean of zero and standard deviation of three. Four chains were run for 1000 iterations each after a burn-in of 500. The median model prediction for each study is shown in Fig. 2a–j with a 95% credible interval, the median predictions from each study are overlaid in Fig. 2k and a pooled prediction of all data is presented in Fig. 2l. There is large variation between studies on the probability of detection by microscopy; at a qPCR-detected parasite density of 100 parasites per μl, the median probability of being detected by microscopy is 29.7%, with a range of 3.8−69.7%.

The proportion of infected individuals in each dataset with parasite densities above certain thresholds (1, 10 and 100 parasites per μl) is shown in Fig. 3. Across all datasets, a mean of 42% (range 1–97%) of infected individuals had densities above 100 parasites per μl, and the proportion of infected individuals detected at this threshold increased in higher prevalence settings. This is consistent with Fig. 1d showing that a lower proportion of infected individuals have subpatent infections in higher transmission settings. By contrast, an average of 56.5% infected individuals (11–100%) had densities above 10 parasites per μl, and the parasite density distributions suggest that a diagnostic tool with this sensitivity would detect >55% of infections in the majority of settings. The proportion of infections that would be detected with a diagnostic with this sensitivity was lower in low transmission settings. For simplicity, this estimate assumes absolute detection thresholds rather than a gradual decrease in sensitivity as parasite density decreases. More than 80% of infected individuals (average of 89.2%) had densities >1 parasite per μl in all settings except those in very low transmission settings (6 out of 10 of the datasets with PCR prevalence <4%). There is a clearer decline in diagnostic performance (i.e., proportion of the infected population detected) in low transmission settings for a threshold of 100 parasites per μl compared with 10 or 1 parasite per μl. For the higher threshold of 100 parasites per μl, 56% of infected individuals are predicted to be detected in the highest transmission, whereas only around 10% would be detected in the very-low-transmission settings (PCR prevalence < 5%). The comparable reduction in sensitivity is from 68% to 29% for a limit of detection (LOD) of 10 parasites per μl and from 93% to 69%

for a LOD of 1 parasite per μl. This could indicate that the reduced sensitivity of microscopy/RDT observed in low transmission settings (Fig. 1d) may be less severe with more sensitive diagnostics.

The results are compared with the actual sensitivity of a new ultra-sensitive rapid diagnostic test (U-RDT) that has been evaluated in cross-sectional prevalence surveys in Myanmar and Uganda[12]. In Uganda, 84% of qPCR-positive infected individuals were also detectable by U-RDT, and in Myanmar this figure was 44%. These values fall between the proportion of the populations detected assuming limits of detection of 1 and 10 parasites per μl for each setting; however, further data are needed to better evaluate the sensitivity of this new test.

**Sensitivity of diagnostics over the course of an infection**. A systematic review was conducted to identify studies which had information on how the detectability by microscopy/RDT of naturally acquired *P. falciparum* parasites changes over time (see the Methods section). We analysed two types of studies that are informative for this question: cohort studies testing individuals by both microscopy and PCR, and studies that monitored population prevalence during a change in transmission intensity.

We identified and requested access to 16 cohort datasets in endemic areas that used both microscopy and PCR. We analysed seven cohorts to which we were granted access within the time frame of the study or which contained the full information in the publication (Table 1). These were all conducted in moderate-to-high transmission areas and the total duration of follow-up was from 3 days to 16 months. Example patterns of slide-negative and slide-positive infection in individuals over time are shown in Fig. 4. Only two of these studies included genotyping infections over time; these found that almost all individuals had multiple clones. Since standard PCR for infection multiplicity does not assess the relative density of different parasite clones, no analyses could be performed on clone-specific density fluctuations. Previous studies have analysed the duration of infection with specific parasite clones over time[32,33]. The cohort studies varied in terms of frequency of follow-up, rates of treatment during the study and age of participants (Table 1 and Supplementary Table 2), but nonetheless show some dynamics of parasitaemia, which are informative about varied endemic settings. Here, we analysed three metrics on the dynamics of infections detected by microscopy/RDT and PCR over time (Table 1), whilst bearing in mind that we do not distinguish new infections over the course of follow-up in these data:

(i) What percentage of infected individuals never test slide-positive during the study?
(ii) What percentage of submicroscopic episodes in individuals are preceded or succeeded by slide-positive samples, further submicroscopic samples or negative samples?

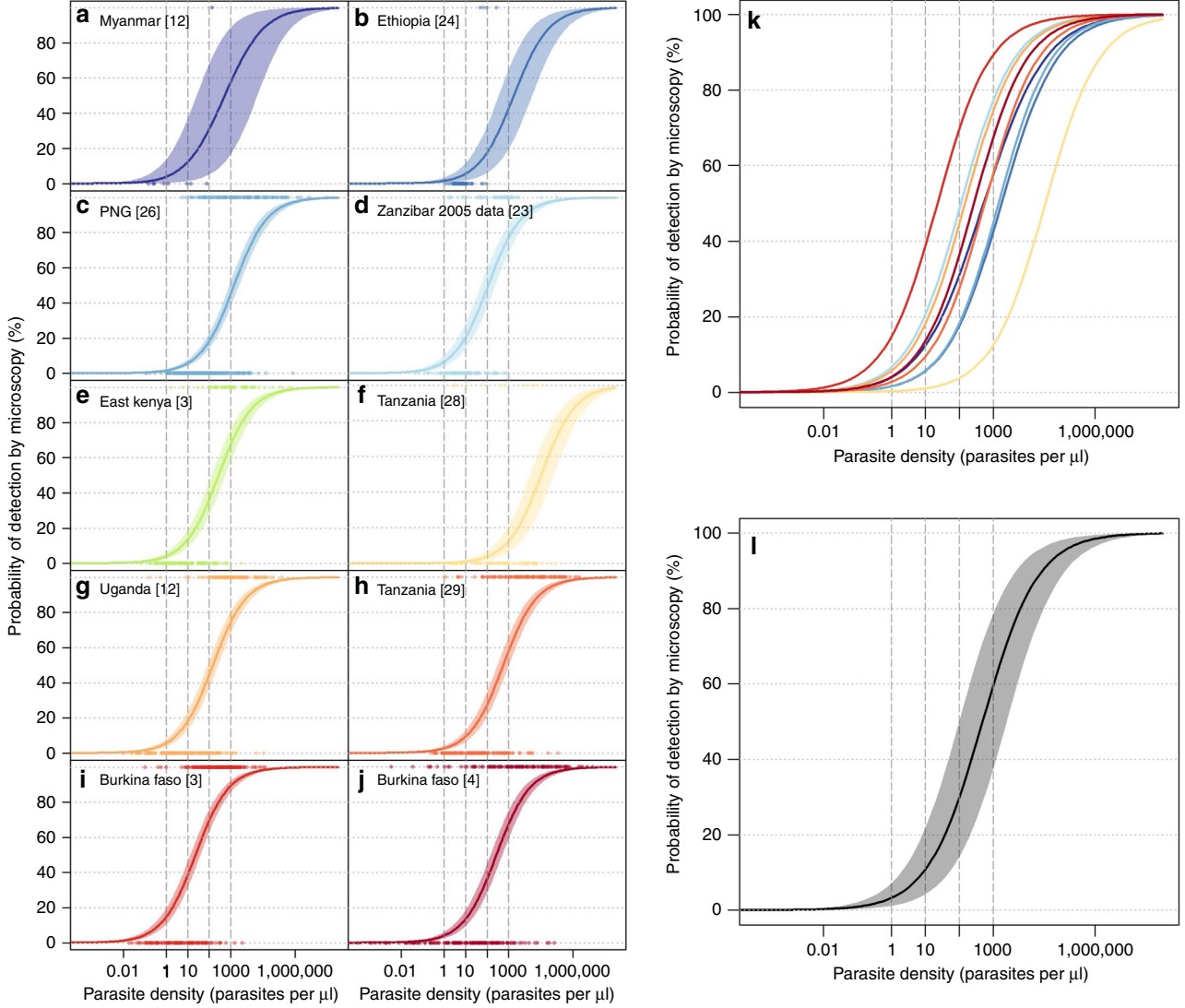

**Fig. 2** Estimated probability of detection by microscopy based on qPCR parasite density. Logistic regression model fits for ten study sites (**a–j**) with associated Bayesian 95% credible intervals (shaded area). Median model predictions for each dataset are presented in panel **k** and a pooled prediction (without the study-level random effect) is presented in panel **l**. Source data are provided as a Source Data file

(iii) What is the relative risk of becoming slide-positive in the future in individuals with submicroscopic infections versus negative individuals?

In each cohort, 0–36% of total samples were submicroscopic, a relatively low level compared with the average of 46% found in a previous systematic review[1]. In all studies, infected individuals were rarely submicroscopic for more than one consecutive sample, although sampling frequency varied from repeated measures < 24 h apart to sampling once every 2 months (Table 1). A relatively large proportion of individuals with submicroscopic infections had negative samples in preceding and subsequent surveys (this occurred in at least 30% of submicroscopic episodes in five of the cohorts). These results could indicate short-lived infections that never became slide-positive, or infections that fluctuated below the PCR detection limit[33–35], or alternatively, slide-positive periods may have been missed between follow-up visits. Most individuals in the cohorts who had any infection during the study tested slide positive on one or more sample, with 0.8% to 18% having submicroscopic-only samples (Table 1). Rates of treatment were higher in the cohort in Papua New Guinea than

in other cohort studies with available treatment data (Supplementary Table 2), but there was no obvious difference in submicroscopic patterns in this cohort.

In most cohorts, current submicroscopic infection was associated with a higher risk of being slide-positive later during follow-up, compared with individuals who were currently negative by PCR (Fig. 5, Table 1). The association was not significant in all studies, but the trend was consistent in 4/5 cohorts where there were sufficient numbers of submicroscopic infections to assess this association. We neither pooled the odds ratios of future slide-positive infection, nor looked at covariates within studies, because the duration and frequency of follow-up was different for each study. Future slide-positive infection could arise directly from the current submicroscopic infection increasing in density, from variable sensitivity of microscopy or could simply be a marker of an exposed individual who is more likely to contract a future infection. The short-term changes in slide-positivity after the initial sample are most likely to be due to either fluctuating densities of the current infection or variable microscopy sensitivity, since new infections would take a longer time to accumulate. One study in Papua New Guinea sampled

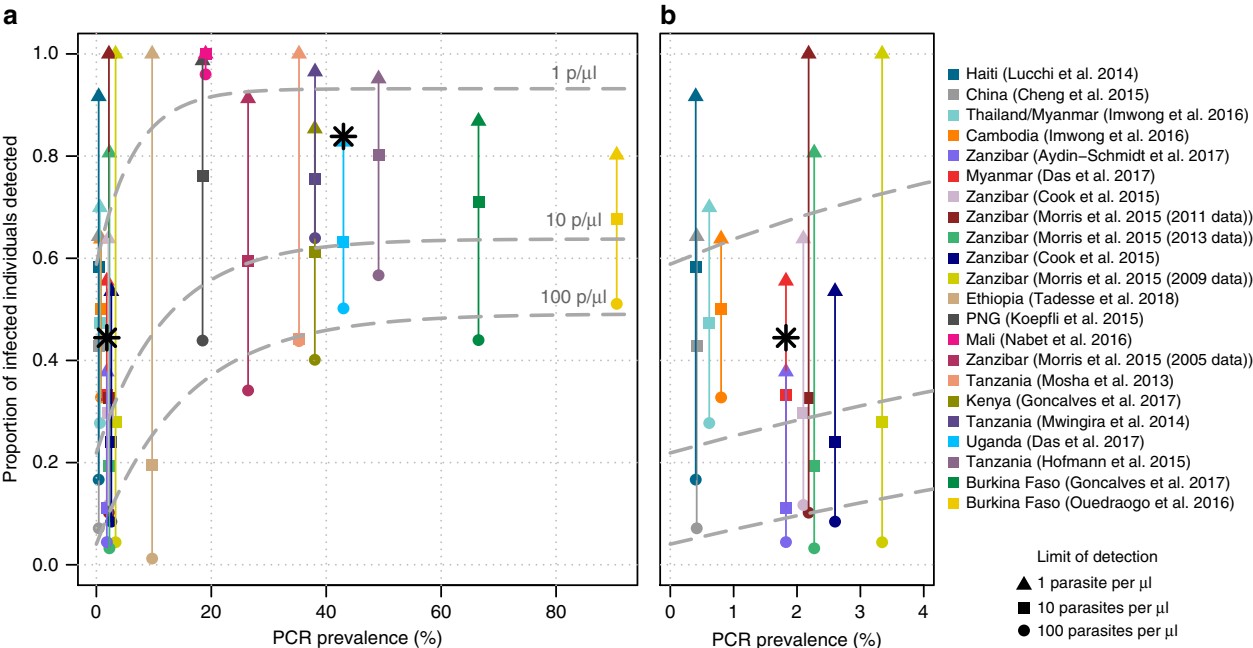

**Fig. 3** The proportion of PCR-positive infections with parasite densities above three thresholds. **a** all datasets— the triangle, square and circle represent the proportion of individuals in each dataset with parasites densities greater than 1, 10 and 100 parasites per µl, respectively (assuming 100% detection below the threshold and 0% detection above, for simplicity). **b** Zoom-in of the 0–4% prevalence area. The black stars indicate the actual proportion of infections detected using the ultra-sensitive RDT in Myanmar (left) and Uganda (right)[12]. The dashed grey lines show the best-fit line (of the form: proportion detected $= a - b * \exp(m * \text{PCR prevalence})$ for the estimated proportion of PCR-positive individuals that would be detected using the three limits of detection. The details of the fitting and the parameter values for the three lines are shown in Supplementary Note 2. Source data are provided as a Source Data file

individuals on consecutive days, showing that about 20% of those initially submicroscopic at day 0 became slide-positive on the day after (Fig. 5). The persistent difference in risk of slide-positive infection by initial infection status which occurred in most cohorts throughout follow-up, up to 35 weeks (Fig. 5), is less likely to be due to the initial infection (most of which would have cleared by this time point). The cause of this pattern is more likely to be variation in individual exposure, or some other long-term difference between individuals (e.g., immunity suppressing parasite densities).

Surveys from multiple time points in areas with seasonal variation in malaria transmission provide an indirect way to look at changing detectability of infections with duration. During the high-transmission season, a larger proportion of infections will have been recently acquired, whilst during dry/low transmission seasons, the average age of infections (the time since acquisition of the most recent parasite clone) will be older, assuming that at least some infections have a duration of several months. Likewise in areas with declining transmission, fewer infections are recent at later time points.

We searched the literature for all cross-sectional surveys that measured *Pf* prevalence by microscopy and a molecular method in different seasons in the same location and by the same authors and laboratory methods (unless otherwise indicated), with the same inclusion criteria in each survey (age etc.) (see the Methods section for search strategy). We included surveys of individuals from defined endemic areas, who were not selected based on malaria symptoms or test results, and excluded surveys that used RDT instead of microscopy, since RDT positivity persists after infection is cleared. We identified 20 locations with two or more surveys, giving a total of 48 surveys (10 locations were identified from a previous review[36] and we found 10 more

locations during the current review). A previous analysis found a non-significant increase in detectability in the rainy season[36]. Here, we analyse the seasonal patterns in more detail with the increased number of datasets. If more than one survey was conducted during the same season and same year in a location, these were pooled. We discarded data from three locations, where the relative difference in slide prevalence between seasons was <5%. Surveys do not necessarily include the same individuals at each time point, therefore this is an ecological analysis.

The sensitivity of microscopy compared with PCR as gold standard was not significantly different between seasons in 17/20 locations (Supplementary Figure 1). When all the seasonal survey results were pooled by meta-regression as previously,[36] the ratio of microscopy sensitivity in the high season relative to the low season was 1.07, 95% CI 0.50–2.31 (i.e., a similar fraction of PCR-positive infections was detected by microscopy in the wet season). We further categorised which locations had high seasonal variation in transmission, defined as the slide-prevalence in the low season being less than half of the slide prevalence in the high season ($n = 10$ locations). In this subgroup of the most seasonal settings, there was a greater increase in the fraction of infections detected by microscopy in the high season compared with the low season, although this difference in sensitivity was not significant (the ratio of microscopy sensitivity in the high season relative to the low season was 1.25, 95% CI 0.66–2.39). The same conclusion was reached if we tested across all sites for a linear effect of the degree of seasonal variation (Supplementary Figure 1) on microscopy sensitivity, rather than using the cut-off to identify the most seasonal settings.

Our literature search also identified five studies in which *Pf* prevalence was measured multiple times by slide/RDT and

**Table 1 Dynamics of infections detected by microscopy/RDT and PCR during longitudinal studies**

| Country, year | Follow-up frequency | Follow-up duration | % sub-microscopic out of infected samples (N) | % of ever infected individuals only having sub-microscopic samples, $n/N^b$ | Odds ratio of any future slide-positive samples in sub-microscopic infected people versus PCR-negatives (95% CI) | Number of consecutive sub-microscopic samples per sub-microscopic episode[b] | Infection status before and after sub-microscopic periods % (n) *[b] |
|---|---|---|---|---|---|---|---|
| Senegal, 2005[62] | 1 week | 9 weeks | 29.7 (277) | 13.0% 22/169 | 2.03 (0.83, 6.48) | 1: 96% (64) 2: 3% (2) 3: 1% (1) | M, SM, M: 5% (2) M, SM, Neg: 25% (10) Neg, SM, M: 5% (2) Neg, SM, Neg: 65% (26) |
| Senegal, 2004[62] | 1 week | 9 weeks | 3.4 (269) | 0.8% 3/369 | N/A | 1: 100% (10) | M, SM, M: 0% M, SM, Neg: 0% Neg, SM, M: 0% Neg, SM, Neg: 100% (4) |
| Papua New Guinea, 2006[65] | 8 weeks, some time points with two surveys 24 h apart | 16 months | 36.1 (1128) | 6.3% 15/239 | 1.72 (0.53, 5.57) | (variable follow-up) | M, SM, M: 30% (105) M, SM, Neg: 25% (87) Neg, SM, M: 16% (58) Neg, SM, Neg: 30% (105) |
| Ghana, 1994–1995[64] | 4 weeks | 12 months | 25.6 (776) | 17.5% 21/120 | 3.13 (1.52, 7.61) | 1: 83% (143) 2: 16% (28) 3: 0.6% (1) | M, SM, M: 22% (22) M, SM, Neg: 8% (8) Neg, SM, M: 19% (19) Neg, SM, Neg: 50% (49) |
| Senegal, 2003[63] [a] | 8 times in 3 days | 3 days | 0.0 (168) | 0% 0/21 | N/A | 0 | M, SM, M: 0% M, SM, Neg: 0% Neg, SM, M: 0% Neg, SM, Neg: 0% |
| Ghana, 2000[32] | 2 months | 12 months | 27.4 (1436) | 9.1% 30/328 | 1.0 (0.36, 2.77) | 1: 81% (116) 2: 22% (25) 3: 1% (2) | M, SM, M: 62% (89) M, SM, Neg: 17% (25) Neg, SM, M: 7% (10) Neg, SM, Neg: 13% (19) |
| Tanzania 1996[66] | 1 month | 6 months | 20.7 (338) | 8% 5/60 | 3.09 (0.92, 10.36) | 1: 76% (52) 2: 18% (12) 3: 4% (3) 4: 0% (0) 5: 1% (1) | M, SM, M: 40% (27) M, SM, Neg: 24% (16) Neg, SM, M: 6% (4) Neg, SM, Neg: 30% (20) |

Details of these studies are given in Supplementary Table 2. Note: the relative densities of different parasite genotypes were not measured, and individuals may have contracted superinfections during the study
*Only includes submicroscopic periods with non-missing samples before and afterwards.
†M, SM, M = microscopy-positive sample, followed by submicroscopic sample(s), followed by microscopy-positive sample
M, SM, Neg = microscopy-positive sample, followed by submicroscopic sample(s), followed by sample negative by PCR and microscopy
Neg, SM, M = sample negative by PCR and microscopy, followed by submicroscopic sample(s), followed by microscopy-positive sample
Neg, SM, Neg = sample negative by PCR and microscopy, followed by submicroscopic sample(s), followed by sample negative by PCR and microscopy
aIndividuals were selected based on being slide-positive at the initial time point
bResults from different studies are not fully equivalent since they depend on sampling frequency, the sensitivity of the different methods, and treatment

molecular methods in cross-sectional surveys by the same authors during a period of declining transmission (Table 2). All areas started with slide prevalence < 10% and reached a slide prevalence of < 1% by the end of the time period. There was always a slightly higher proportion of subpatent infections in the last survey (6–27% increase in subpatent infections), despite fluctuation in sensitivity during the studies. Although RDT positivity can persist after an infection, its duration (up to 6 weeks[37]) is much shorter than the minimum time gap between surveys here (6 months), so persistence would be unlikely to affect results.

**Gametocyte carriage and infectivity.** Data on gametocyte prevalence and density, measured by a PCR in asymptomatically infected individuals, were taken from the studies identified in the first section[3,4,26,28,38,39] plus one additional study[39] (Supplementary Table 3). The PCR prevalence of gametocytes was significantly higher ($p < 0.05$, Pearsons Chi-squared test) in individuals with patent asexual infections compared with subpatent asexual infections in six of the seven studies (Fig. 6). The proportion of subpatent individuals with PCR-detectable gametocytes increases in higher transmission settings (OR:1.05 (1.04–1.07) $p < 0.05$, Wald test), but the same is not true for individuals with patent infection ($p > 0.05$, Wald test). Although

subpatent infections are associated with lower densities of both asexual parasites and gametocytes, the distribution of gametocyte density of subpatent individuals was significantly different ($p < 0.05$, Kolmogorov–Smirnoff test) from than the densities of patent individuals in only three of the seven studies.

The contribution to the infectious reservoir (the combined onwards infectivity of a population to mosquitoes) of individuals with subpatent infection was calculated for several of the studies where mosquito feeding was conducted[3–5,40] (details in Supplementary Table 4). Using the methodology outlined in Stone et al.[41] where the relative contribution of each individual to the infectious reservoir depends on the relative proportion of mosquitoes they infected, weighted to ensure the age distribution of the population is equal for each study (15% under 5 years old, 30% 5–15 and 55% over 15), the relative bites received by each individual are a function of their body surface area and how likely they are to use a bednet. Figure 7 shows the contribution to the infectious reservoir of individuals with subpatent infection (light blue area in fourth bar in each panel). The studies, ordered by PCR prevalence, are all from moderate and high transmission settings. The contribution to the infectious reservoir of all individuals with subpatent infections is greater in moderate compared with high transmission settings. The robustness of these results is dependent on the sample sizes in the studies, in

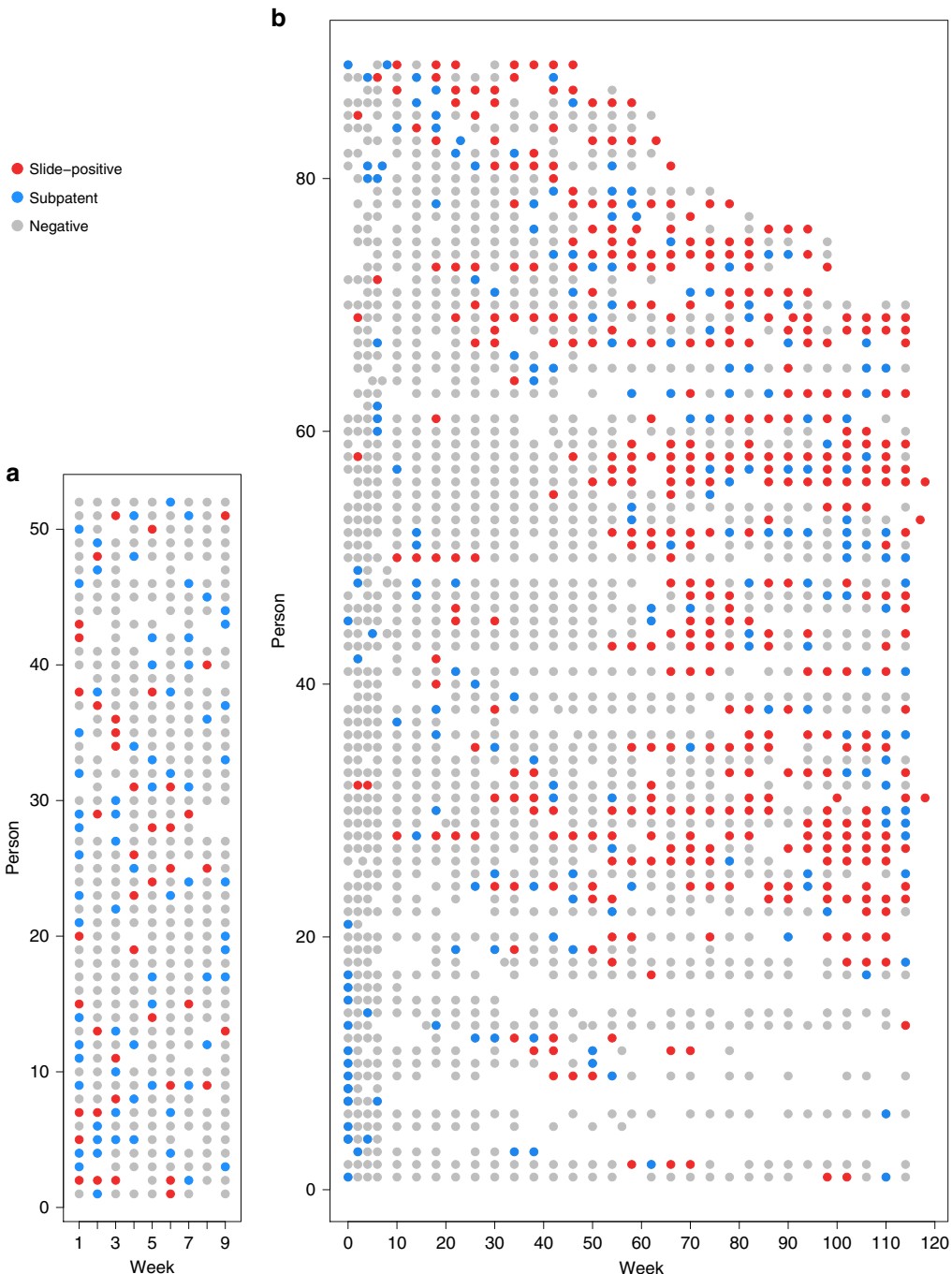

**Fig. 4** Example longitudinal data in a subset of individuals who experienced submicroscopic parasitaemia. The data include information by microscopy (slide-positive) and PCR in (**a**) all-age individuals in Senegal in 2005[62] and (**b**) a birth cohort in Ghana[64]. Each row represents a single individual, with time on the x-axis (time = age in the Ghana cohort). Individuals were sampled weekly in Senegal and every 2–4 weeks in Ghana, and the colours indicate their infection status at each sampling time. Blank space indicates missing data. Source data are provided as a Source Data file

particular the number of mosquitoes that were infected in each study. Two of the locations in Fig. 6 had < 20 mosquitoes getting infected (Kilifi: 4/10,763, Mbita: 16/13,913), meaning these estimates have higher uncertainty.

In Mbita (where only microscopy data were available), an additional 17 mosquitoes were infected from one individual who was negative for asexual *falciparum* parasites and positive for *falciparum* gametocytes (at a very low density). This person was additionally positive for *malariae* asexual parasites and negative for *malariae* gametocytes. Therefore it is plausible

that these 17 infected mosquitoes could be infected with *falciparum*, *malariae* or a combination. Here, we have adopted a conservative assumption that they were infected with *malariae*, but if we were to assume they were infected with *falciparum*, as this individual is classified as subpatent (based on their asexual *P. falciparum* parasitaemia), this would increase the contribution to the infectious reservoir of subpatent individuals from 19.5% (shown here) to 55.4%.

The relative infectiousness to mosquitoes of individuals with microscopy-detectable infections compared with individuals

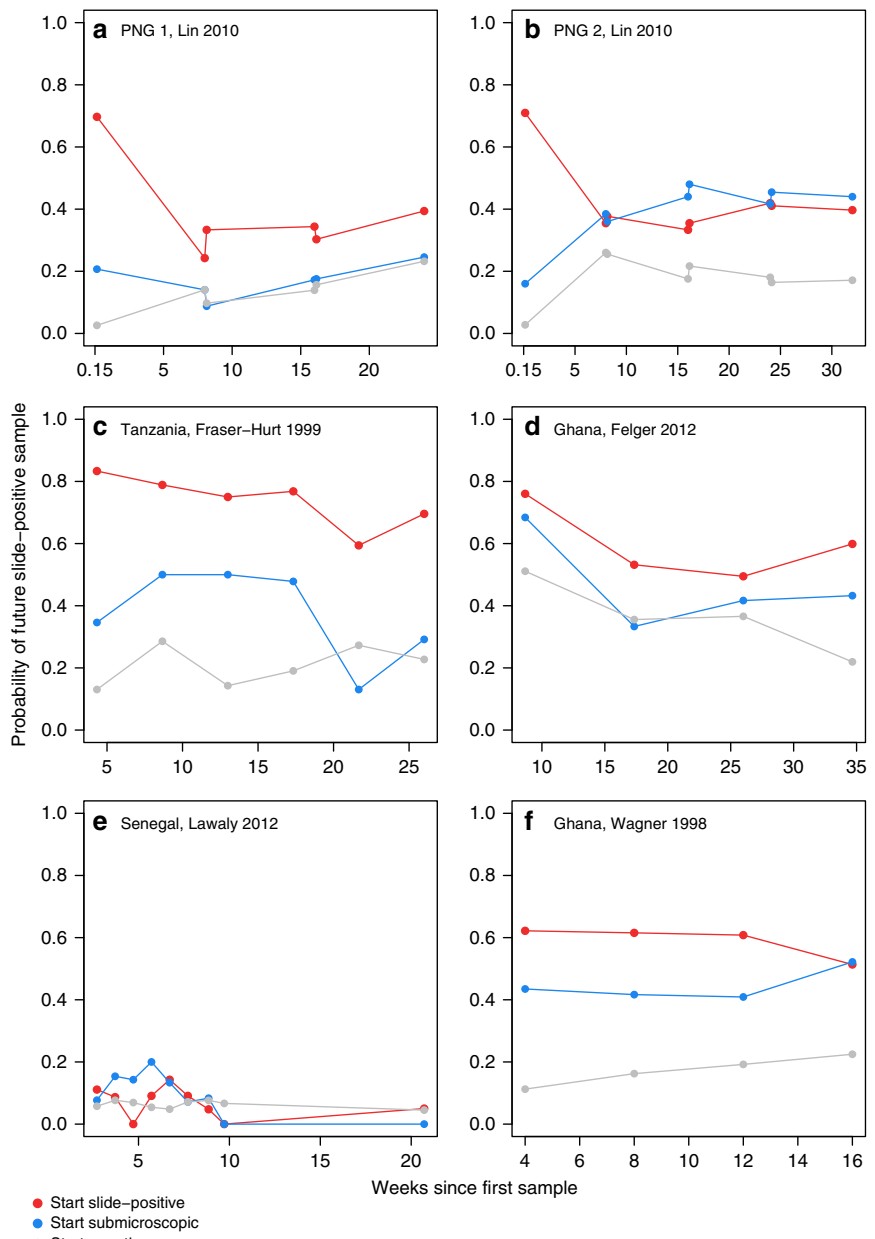

**Fig. 5** The probability of being slide-positive after the current sample, by initial infection status. Data are shown for the cohorts listed in Table 1 which had available data. Country, first author and year of publication are indicated. We excluded cohorts in which no individuals experienced submicroscopic infection. The Papua New Guinea (PNG) cohort was divided into two because of its long duration, and to show that the choice of start point can be important (PNG 1 started May 2006 and PNG 2 started November 2006). The PNG cohort had a first follow-up time on day 1 and afterwards was followed up on consecutive days every 8 weeks (top 2 panels). Source data are provided as a Source Data file

with submicroscopic infections was compared across eight studies[3–5,24,39,40,42] (Fig. 8) (details in Supplementary Table 4). Individuals with microscopy/RDT detectable infections were more infectious in all studies except one which was conducted in Senegal (which had a relative infectiousness of 0.99 (CI: 0.54–1.79)). Overall, the pooled relative infectiousness of individuals with microscopy or RDT detectable infections was 2.87 times greater (95% CI: 2.54–3.25) than individuals with subpatent infections.

### Discussion
The proliferation of quantitative molecular methods has allowed us to take a fresh look at the parasite densities, detectability

and dynamics of low-density *Plasmodium falciparum* infections. We have shown that infected individuals have, on average, lower parasite densities in lower transmission settings, but in a given setting, the range of parasite densities in infected individuals can span several log orders of magnitude. Several hypotheses have been proposed to explain why parasite densities are lower in low transmission settings, and the answer is likely to be a complex and interacting combination of the following factors. Firstly, this may simply be because individuals receive fewer infectious bites and are therefore on average further along in their course of infection where parasitaemia is expected to be lower[1,43]. Additionally, populations in areas that are now low transmission but were higher transmission in the past will still have acquired

**Table 2 Sensitivity of microscopy or RDT compared with PCR in areas with declining transmission**

| Country | Year | Microscopy or RDT | Microscopy/ RDT prevalence % (N) | PCR prevalence % (N) | Sensitivity of microscopy/RDT | Reference |
|---|---|---|---|---|---|---|
| Brazil* | 2004 (Mar–Apr) | Microscopy | 1.5 (388) | 9.1 (386) | 16.5 | 70 |
| | 2004 (Sep-Oct) | Microscopy | 1.6 (378) | 8.7 (379 | 18.4 | |
| | 2005 | Microscopy | 0.0 (329) | 6.7 (328) | 0 | |
| | 2006 | Microscopy | 0.3 (351) | 2.4 (334) | 12.5 | |
| Kenya | 2012 | Microscopy | 2.0 (779) | 6.2 (779) | 32.3 | 68 |
| | 2013 | Microscopy | 0.2 (797) | 3.3 (797) | 6.1 | |
| Tanzania | 2005 | Microscopy | 1.9 (2721) | 32.5 (453) | 5.8 | 69 |
| | 2008 | Microscopy | 0.0 (370) | 2.8 (145) | 0 | |
| Tanzania, Zanzibar† | 2005 | Microscopy | 7.5 (2471) | 21.1 (534) | 35.5 | 23 |
| | 2009 | Microscopy | 0.0 (2423) | 3.3 (2423) | 0 | |
| | 2011 | RDT | 0.4 (2904) | 2.2 (2977) | 18.2 | |
| | 2013 | RDT | 0.3 (3026) | 2.3 (3038) | 13 | |
| Zambia† | 2009 | RDT | 0.7 (676) | 2.7 (638) | 25.9 | 67 |
| | 2010 | RDT | 0.2 (871) | 1.8 (871) | 11.1 | |
| | 2011 | RDT | 0.4 (740) | 1.5 (740) | 26.7 | |
| | 2012 | RDT | 0 (688) | 0.4 (688) | 0 | |

Sensitivity is measured by slide and PCR unless otherwise indicated: Zanzibar[23] (microscopy used up to 2009, RDT used > 2009), Zambia[67] (RDT), Kenya[68], Tanzania[69] (QT-NASBA), Brazil[70]
*Blood samples were finger prick in 2004 and 2006 and venous in 2005
†Filter papers in earlier years were stored for longer prior to PCR

immunity, therefore may be better able to control parasite density than expected based on the current level of malaria exposure. Individuals in areas that have historically low transmission would not be expected to have much acquired immunity, however, may harbour low parasite densities because they reside in small geographic pockets of high transmission where immunity is higher[44]. Another contributing factor could be that the low genetic diversity of parasite populations in low transmission settings enables individuals to rapidly acquire immunity to those parasite clones[45,46].

The decrease in average parasite density as an infection progresses, so that it is more often below the detection threshold of microscopy, is clearly seen in artificially induced untreated malariatherapy infections[1], but in our analyses of cohort studies, we could not test for this due to frequent reinfection. However, there was a non-significant trend towards higher sensitivity of microscopy in the rainy season when infections would be expected to be more recent and potentially more detectable, compared with the dry season.

The wealth of quantitative PCR data collated for this study allowed us, for the first time, to investigate how the proportion of infections detected in different transmission settings might be expected to change with more sensitive diagnostics. We estimate that the proportion of PCR-positive individuals detected with a diagnostic with a LOD of 100 parasites per μl (akin to microscopy/RDT) decreases from 49% in medium–high transmission settings (> 10% PCR prevalence) to only 14% in low transmission settings (< 10% PCR prevalence). With a more sensitive diagnostic, the expected drop-off in performance is smaller. The estimated proportion of individuals detected decreased from 63% in medium–high transmission settings to 33.1% in low transmission settings when the LOD is 10 parasites per μl, and from 94% to 74% when the LOD is one parasite per μl. This suggests that more sensitive point of care diagnostics (such as LAMP or a more highly sensitive RDT) are essential for detecting asymptomatic individuals in low transmission settings. A caveat of this analysis is potential variation in results between laboratories and experiments. While the sensitivity of many PCR techniques for detection of presence of malaria appear comparable[47], variation in results is particularly an issue for quantitative measures of

parasitaemia. One study found a 2.4-fold variation in estimated densities between replicates in the same laboratory[48], and sensitivity has been shown to vary based on the method used[29,49]. However, the large number of studies included in our analysis and the relative consistency of the trend towards higher parasite densities in higher transmission settings increases the robustness of these findings.

In our analysis of cohort studies, submicroscopic infection was a predictor of future microscopy-detectable infection in four out of five cohorts. Understanding whether this is due to an increase in the parasite density of the current submicroscopic infection, or because a submicroscopic infection is a marker of higher exposure and risk of future infection, has important implications for the impact of treating submicroscopic infections. Treatment could prevent a current infection becoming higher density and more infectious, but would not protect an individual with higher exposure getting infected in the future after the prophylactic effect of the antimalarial has waned. Recent cohort studies in Vietnam and Mozambique found that some individuals with untreated low-density *P. falciparum* infections later had high density infections[50,51]. While the possibility of new infections could not be excluded in these studies due to low levels of DNA for genotyping the submicroscopic infections, the Vietnam study was in a very low transmission area with low reinfection risk, suggesting the lower density infection later increased in density.

We did not directly estimate the duration of submicroscopic infection in this analysis due to infrequent follow-up times during cohort studies, and the presence of multiple clone infections. However, the mean total duration of infection with a particular parasite genotype has been estimated at around 6 months (ranging from a few weeks to several years) in previous analyses in high transmission areas[8,12,33]. In lower transmission areas, a shorter duration with a median of 2 months or less has been estimated using ultra-sensitive PCR[50,52], but this was in cohorts with frequent treatment and also it was not possible to exclude continued infection below the limit of PCR detection as has been seen in other cohorts by genotyping[33]. Wider application of quantitative molecular techniques in future is likely to provide further insight into the fluctuations of clone-specific densities in endemic populations (e.g., ref. [53]).

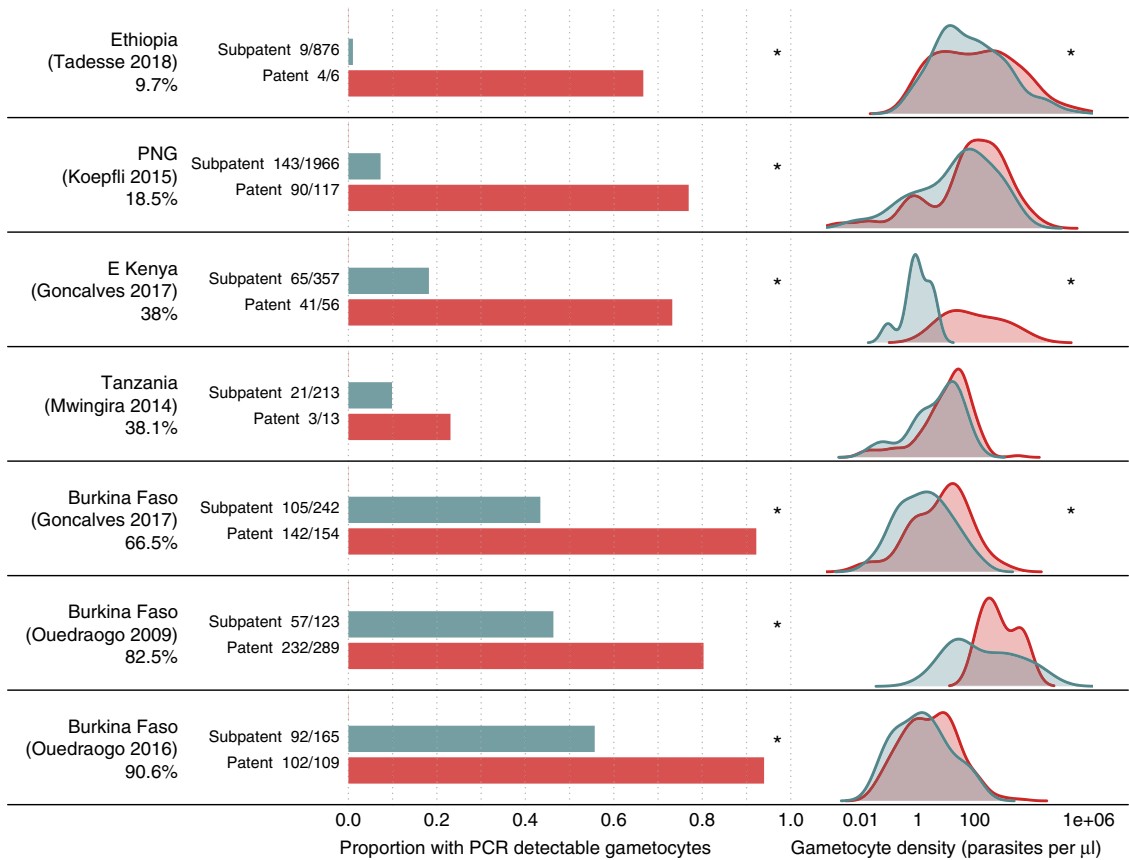

**Fig. 6** Prevalence and density of gametocytes. Gametocyte prevalence and density of gametocytes in individuals with patent and subpatent asexual Plasmodium falciparum infections. The studies are ordered by community asexual prevalence (shown by the percentage under the study name), with the lowest at the top. The density plots show the gametocyte densities (by PCR) of individuals with patent asexual (red) and subpatent asexual (blue) infections. *Indicates statistical significance for the two tests described in the methods. There is poor comparability of gametocyte densities between different laboratories, therefore the plots in the right column should be used to compare between the patent and subpatent parasite density distributions in each study, rather than distributions between studies. Source data are provided as a Source Data file

Low-density infections have repeatedly been shown to be infectious[4,54], although their relevance to onwards transmission may depend on local vector competence[55] and the likelihood that infected mosquitoes produce infectious sporozoites[56,57]. Here, a meta-analysis of eight studies with mosquitoes feeding on asymptomatically infected individuals estimated that on average an individual with subpatent infection is approximately a third (0.348 times) as infectious to mosquitoes as a microscopy-positive individual. As the proportion of infections that are subpatent increases in low transmission settings (Fig. 1d), these individuals are predicted to contribute an increasing proportion to the infectious reservoir of the infected population. In low transmission settings, symptomatic infections may also be an increasingly important contributor to the infectious reservoir, but these infections are often not captured in cross-sectional prevalence surveys. Understanding the relative contribution to the infectious reservoir of symptomatic and patent and sub-patent asymptomatic individuals is key to determining whether these latter groups need to be detected and treated in order to interrupt transmission and achieve local elimination[24]. The relative impact of treating asymptomatic individuals in comparison with other interventions such as improving access to treatment for symptomatic cases or improved vector control remains to be quantified in the field[58]. A recent study in Myanmar showed that combining MDA in higher transmission villages with widespread improved access to

treatment resulted in a marked and sustained reduction in transmission, with no transmission documented in many areas for > 6 months[59,60].

In this study, we used detailed data from a range of transmission settings to pave the way for understanding the relative importance of subpatent malaria infections and assessing whether they need to be targeted specifically in elimination settings. We have shown that infected individuals on average have lower parasite densities in low transmission settings and that these infections are likely to be infectious to mosquitoes in the field. Furthermore, although these infections may have shorter durations and lower infectiousness compared with patent infections, we have shown that submicroscopic infections are predictive of future patent infection. Therefore, detecting and treating these infections may be an effective approach to prevent future periods of parasitaemia and identifying micro 'hotspots' of higher exposure and transmission. We have also shown that a large proportion of infected individuals have parasite densities below the LOD of standard field diagnostics in low transmission settings, and that higher sensitivity field diagnostics are needed if it is decided that detecting and treating these infections is necessary. The next step is to synthesise the results from this analysis with future field data on interventions which target low-density malaria infections to further understand when, where, and whether we need to identify and treat these low-density infections to interrupt transmission.

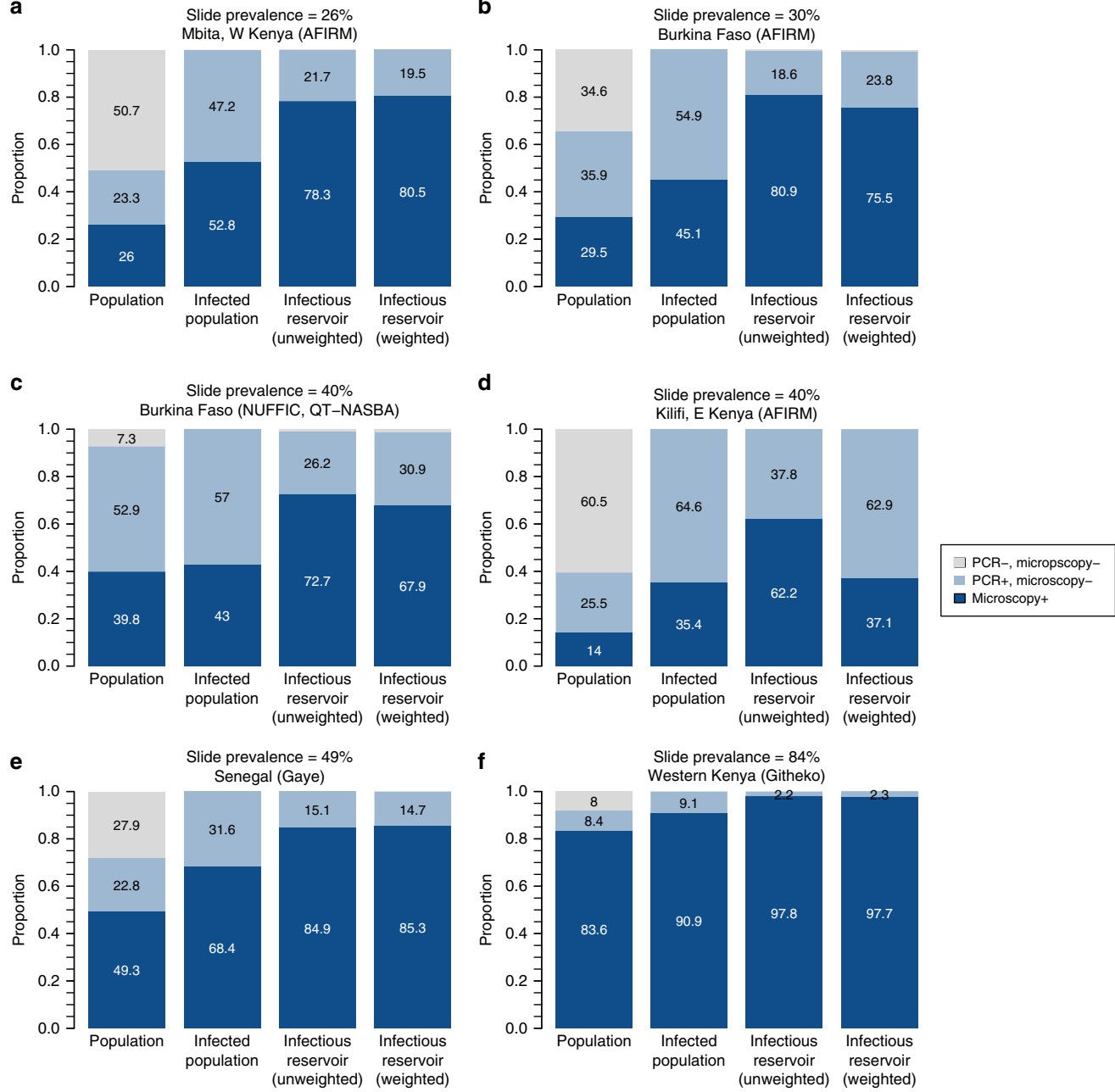

**Fig. 7** Contribution to the infectious reservoir of individuals with microscopic and subpatent infections. The results are shown for data from six study locations[3–5, 40]. The first column in each panel shows the proportion of the population that are infected with microscopy or RDT detectable and undetectable asexual parasites. The second column shows how these two groups make up the infected population. The third column shows the unweighted contribution to the infectious reservoir of the two groups, accounting for the proportion of people in each group and their relative infectivity to mosquitoes. The weighted infectious reservoir shown in the fourth column additionally accounts for the relative biting frequency (based on their age and probability of using a bednet) of individuals and corrects for the age distribution of participants in each study. Source data are provided as a Source Data file

## Methods

**Systematic review**. There are three main topics within our review (see below) and we designed specific search strategies for each. Abstracts of publications within PubMed were searched for the specific terms below. The authors also sought relevant unpublished data as far as possible within the time frame of the review.

1. *Density and detectability of P. falciparum infections:* search after 01/11/2011 (date at which previous systematic review[1] was completed) for:

- falciparum AND (densit* AND (pcr OR polymerase chain reaction)) OR "quantitative pcr" OR "quantitative polymerase chain reaction" OR qpcr OR "real time pcr" OR real time polymerase chain reaction OR NASBA

- falciparum AND (densit* OR quantitative) AND (LAMP OR loop-mediated isothermal amplification OR thermophilic helicase dependent amplification OR tHDA OR recombinase polymerase amplification OR RPA)

2. *Sensitivity of diagnostics over the course of an infection:* search terms were (no date restrictions):

- falciparum AND (within-host OR "within host")
- falciparum AND (longitudinal OR cohort OR repeated OR duration) AND (PCR OR polymerase chain reaction OR LAMP OR loop-mediated isothermal amplification OR thermophilic helicase dependent amplification OR tHDA OR recombinase polymerase amplification OR RPA)

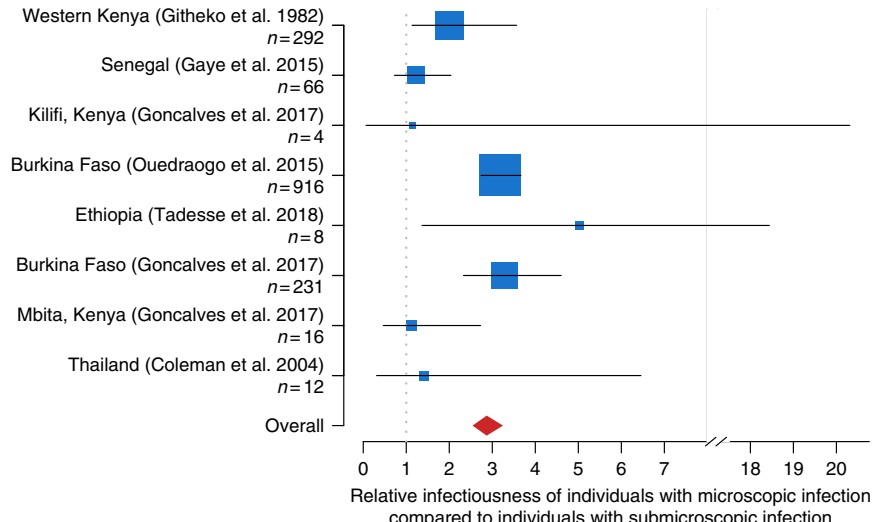

**Fig. 8** Relative infectiousness of submicroscopically infected individuals. Relative infectiousness of individuals with microscopy detectable infection compared to individuals with subpatent infection. The location of the blue box shows the mean, the size of the blue box indicates the number of infected vectors in the study (also shown in the y-axis labels). The horizontal lines indicate the 95% confidence intervals and the red diamond shows the pooled mean. Studies are listed in order of prevalence by microscopy, from highest at the top to lowest at the bottom. Data from each study are weighted to account for differences in expected biting frequency based on age (due to body size and probability of using a bednet) and age groups are weighted to ensure a consistent age distribution of individuals in all studies. Source data are provided as a Source Data file

3. *Gametocyte carriage and infectivity*: search terms (no date restrictions)

- falciparum AND reservoir AND (infectivity OR infectious)
- falciparum AND infectivity AND (gametocyte OR submicroscopic OR asymptomatic)
- falciparum AND (gametocyte density OR gametocytemia) AND transmission
- vectorial capacity AND (gametocytaemia OR gametocyte).

**Statistical analysis**. The relationship between PCR parasite density and detectability by microscopy was modelled using a logistic regression model with a study-level random effect fitted using a Bayesian framework and Stan[61]. For the analysis of cohort studies, to test whether individuals with submicroscopic infections are more likely than PCR-negative individuals to become slide positive in the future, we computed the probability of slide-positive infection at any follow-up time, by initial infection status. We calculated the odds ratio and 95% confidence intervals of future slide-positive samples using a continuity correction and small sample size correction since the number of individuals in the initially submicroscopic group was sometimes small. One cohort in Senegal in 2004[62] and one in 2003[63] were not included, as there were too few individuals with submicroscopic infection. One cohort of 0–2 years old in Ghana[64] had only a few individuals who were initially submicroscopic. Here, we subdivided the follow-up samples into five 16-week intervals beginning at 6 months of age (excluding earlier time periods because of maternal immunity), and took the initial infection status at the start of each time. The follow-up frequency was the same in the different time periods. The risk of slide-positive infection was then computed for each follow-up period using logistic regression with random subject effects to account for repeated measures.

The proportions of patent versus subpatent individuals with PCR-detectable gametocytes was compared using a chi-squared test. The relationship between community level (asexual parasite) PCR prevalence and the proportion of patent and subpatent individuals with PCR-detectable gametocytes as assessed using a generalised linear mixed effects model with asexual parasite prevalence as a fixed effect and study as a random effect. The distribution of parasite densities of gametocytes in patent versus subpatent individuals was compared using a two-sided, two-sample Kolmogorov–Smirnoff test on (log10) transformed data.

**Reporting Summary**. Further information on experimental design is available in the Nature Research Reporting Summary linked to this article.

**Disclaimer**. The findings and conclusions in this report are those of the author(s) and do not necessarily represent the official position of the Centers for Disease Control and Prevention.

## Data availability

All data used in this manuscript are provided as Source Data files, which are also deposited on dryad (https://doi.org/10.5061/dryad.75t5382), or in online repositories previously deposited by study principle investigator for each relevant study with original publication of data. Details of repositories are provided in the Source Data files on tabs relating to the relevant study. All data supporting the findings of this study are also available from the authors upon request.

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

## Acknowledgements

H.C.S. would like to acknowledge funding support from an Imperial College Junior Research Fellowship. L.C.O. acknowledges funding from a UK Royal Society Dorothy Hodgkin fellowship, the Bill & Melinda Gates Foundation and Medicines for Malaria Venture. R.Pa. would like to acknowledge funding from the Strategic Anopheles Horizontal Research Programme, Institut Pasteur. H.C.S. and L.C.O. acknowledge joint Centre funding from the UK Medical Research Council and Department for International Development.

## Author contributions

H.C.S. and L.C.O. drafted the paper. H.C.S., L.C.O., T.B. and C.D. conceived the study. H.C.S., L.C.O. and A.R. designed and conducted the analyses. I.F., N.H., L.R., J.C., B.P.G., A.B., A.L.O., U.M., M.M., C.K., I.M., F.T., E.G., S.M., G.D., M.K., J.Mi., S.O.A., C.N., R.Pi., O.D., S.N.D., K.K., N.L., V.U., J.Mo., A.T., D.C., R.G., F.M., R.S., R.Pa., E.M.R., N.J.W., F.N., M.I., T.B. and C.D. collected the data. All authors contributed to interpretation of the analyses and revised the draft paper.

## Additional information

**Competing interests:** The authors declare no competing interests.

Hannah C. Slater[1], Amanda Ross[2,3], Ingrid Felger[3,4], Natalie E. Hofmann[3,4], Leanne Robinson[5,6,7,8], Jackie Cook[9], Bronner P. Gonçalves[10], Anders Björkman[11], Andre Lin Ouedraogo[12,13], Ulrika Morris[11], Mwinyi Msellem[14], Cristian Koepfli[15,16], Ivo Mueller[6,17,18], Fitsum Tadesse[19,20,21], Endalamaw Gadisa[20], Smita Das[22], Gonzalo Domingo[22], Melissa Kapulu[23,24], Janet Midega[23,24], Seth Owusu-Agyei[25], Cécile Nabet[26], Renaud Piarroux[26], Ogobara Doumbo[27], Safiatou Niare Doumbo[27], Kwadwo Koram[28], Naomi Lucchi[29], Venkatachalam Udhayakumar[29], Jacklin Mosha[30], Alfred Tiono[31], Daniel Chandramohan[32], Roly Gosling[33], Felista Mwingira[34], Robert Sauerwein[19], Richard Paul[35], Eleanor M Riley[10,36], Nicholas J White[37,38], Francois Nosten[37,39], Mallika Imwong[38,40], Teun Bousema[10,19], Chris Drakeley[10] & Lucy C Okell[1]

[1]MRC Centre for Global Infectious Disease Analysis, Department of Infectious Disease Epidemiology, Imperial College London, London W2 1PG, UK. [2]Department of Epidemiology and Public Health, Swiss Tropical and Public Health Institute, Basel 4002, Switzerland. [3]University of Basel, Basel 4001, Switzerland. [4]Medical Parasitology and Infection Biology, Swiss Tropical and Public Health Institute, Basel 4002, Switzerland. [5]Vector-borne Diseases Unit, Papua New Guinea Institute for Medical Research, Madang, Papua New Guinea. [6]Division of Population Health and Immunity, The Walter and Eliza Hall Institute of Medical Research, Parkville 3052 VIC, Australia. [7]Medical Biology, University of Melbourne, Melbourne 3010 VIC, Australia. [8]Disease Elimination, Burnet Institute, Melbourne 3004 VIC, Australia. [9]MRC Tropical Epidemiology Group, London School of Hygiene and Tropical Medicine, London WC1E 7HT, UK. [10]Department of Immunology and Infection, London School of Hygiene and Tropical Medicine, London WC1E 7HT, UK. [11]Malaria Research, Department of Microbiology, Tumor and Cell Biology, Karolinska Institutet, 171 77, Stockholm, Sweden. [12]Département de Sciences Biomédicales, Centre National de Recherche et de Formation sur le Paludisme, Ouagadougou 01 BP 2208, Burkina Faso. [13]Institute for Disease Modeling, Intellectual Ventures, Bellevue 98005 Washington, USA. [14]Department of Training and Research, Mnazi Mmoja Hospital, Zanzibar, Tanzania. [15]Population Health and Immunity Division, Walter and Eliza Hall Institute, Melbourne 3052 Victoria, Australia. [16]Department of Biological Sciences, University of Notre Dame, Indiana 46556, USA. [17]Department of Parasites and Insect Vectors, Institut Pasteur, Paris 75015, France. [18]Medical Biology, University of Melbourne, Melbourne 3010 VIC, Australia. [19]Radboud Institute for Health Sciences, Radboud University Medical Centre, Nijmegen 6525, The Netherlands. [20]Armauer Hansen Research Institute, Addis Ababa, Ethiopia. [21]Institute of Biotechnology, Addis Ababa University, Addis Ababa, Ethiopia. [22]Diagnostics Program, PATH, Seattle, Washington 98121, United States of America. [23]Centre for Tropical Medicine and Global Health, Nuffield Department of Medicine, University of Oxford, Oxford OX3 7FZ, UK. [24]KEMRI–Wellcome Trust Research Programme, Centre for Geographic Medicine Research-Coast, Kilifi, Kenya, Centre for Genomics and Global Health, Wellcome Trust Centre for Human Genetics, University of Oxford, Oxford OX3 7BN, UK. [25]Institute of Health, University of Health and Allied Sciences, Hohoe PMB 31, Ghana. [26]Sorbonne Université, INSERM, Institut Pierre-Louis d'Epidémiologie et de Santé Publique, AP- HP, Groupe

Hospitalier Pitié-Salpêtrière, Service de Parasitologie-Mycologie, Paris 75646, France. [27]Malaria Research and Training Centre, Parasitic Diseases Epidemiology Department, UMI 3189, University of Sciences, Technique and Technology, Bamako, Mali. [28]Noguchi Memorial Institute for Medical Research, University of Ghana, Legon, Ghana. [29]Malaria Branch, Division of Parasitic Diseases and Malaria, Centers for Global Health, Centers for Disease Control and Prevention, Atlanta 30030 GA, United States of America. [30]National Institute for Medical Research, Mwanza Medical Research Centre, Mwanza, Tanzania. [31]Department of Biomedical Sciences, Centre National de Recherche et de Formation sur le Paludisme, Ouagadougou 01 BP 2208, Burkina Faso. [32]Department of Disease Control, London School of Hygiene and Tropical Medicine, Keppel Street, London WC1E 7HT, UK. [33]Malaria Elimination Initiative, Global Health Group, University of California, San Francisco, San Francisco 94158 CA, United States. [34]Biological Sciences Department, Dar es Salaam University College of Education, P. O. Box 2329, Dar es Salaam, Tanzania. [35]Institut Pasteur de Dakar, Laboratoire d'Entomologie Médicale, Dakar, Senegal. [36]The Roslin Institute and Royal (Dick) School of Veterinary Studies, University of Edinburgh, Easter Bush, Midlothian EH25 9RG, UK. [37]Centre for Tropical Medicine and Global Health, Nuffield Department of Clinical Medicine, University of Oxford, Oxford OX3 7FZ, UK. [38]Mahidol Oxford Tropical Medicine Research Unit, Faculty of Tropical Medicine, Mahidol University, Bangkok 10400, Thailand. [39]Shoklo Malaria Research Unit, Mahidol–Oxford Tropical Medicine Research Unit, Faculty of Tropical Medicine, Mahidol University, Mae Sot 63110, Thailand. [40]Department of Molecular Tropical Medicine and Genetics, Faculty of Tropical Medicine, Mahidol University, Bangkok 10400, Thailand. Deceased: Ogobara Doumbo

