## [Peer Review File · Nature Communications]

Reviewers' Comments:

Reviewer #1:

Remarks to the Author:

This is a well written paper on a very important issue in malaria epidemiology, namely the temporal dynamics of asexual stage parasite densities during a *P. falciparum* infection, as well as the infectivity to mosquitoes associated with such varying densities over time. It is particularly important that the researchers point out the deficiency in assessing subpatent infections from cross-sectional studies, which do not allow the assessment of how parasite densities vary over time during an infection. To address this temporal dynamic of infections, the team identified cohort studies with repeated microscopy-PCR measures from a systematic review of the published literature, which to my knowledge represents the first time this has been done in a comprehensive multi study approach. This research is important to malaria epidemiology. The methods and statistical analyses are valid and form a solid foundation from which the conclusions are drawn.

There are a few areas that should be considered to help improve the paper:

- Are there any data in the cohort studies on fever at each follow-up period so that clinical malaria episodes could be estimated and the association with parasite density explored? I'm assuming they would have needed to be treated if they developed symptoms, but this still might provide valuable insight into the natural history of a malaria parasite infection and symptomology.
- In assessing the risk of current submicroscopic infected individuals in the cohorts becoming slide-positive at a later follow-up period, compared to PCR-negative individuals, the time periods being assessed would likely impact this relationship. Can the authors please explain this finding a little further?
- On page 16 (line 1-2), can the authors please explain how they arrived at a ratio of slide prevalence during the low vs. high season as 0.5 to indicate high seasonality? This seems arbitrary. An alternative approach to this analysis would be to include all studies in a simple logistic regression model for the odds ratio of wet vs. dry season on slide prevalence, while including a covariate for highly season vs not.
- The likely reasons are stated as to why individuals in low transmission areas have lower parasite densities, as compared to their counterparts in higher transmission areas. However, it is likely a combination of having few new infectious bites, highly clonal parasite populations, and potentially elevated acquired immunity due to highly heterogeneous transmission over space.
- One of the primary weaknesses of this study is that it was not possible to account for new *Pf* infections among the cohort over time. The authors argue that while parasite densities in the malaria therapy studies showed a clear decrease in parasite densities after the initial peak in asexual stage parasites, their analysis of the cohorts did not support this (lines 1-6 on page 26). However, the malaria therapy data also showed a second peak in asexual stage parasite densities when an infected individual was inoculated with a second new infection, especially with a different parasite strain. This would then suggest that the likely reason for not seeing a stable decrease in parasite densities over time in the cohorts in endemic areas was that repeated reinfection of individuals kept densities at various levels over time.
- Following from the above point, on the observation that submicroscopic infections were a significant predictor of a later microscopy-detectable infection, it appears most likely that the mode of action is that the submicroscopic infection acting is a marker for higher risk of acquiring a new infection in the future, and thus causing a rebound in parasite densities at that time. The cohort study cited to suggest parasite densities vary over time, even with a low chance of acquiring a new infection, is from a *P vivax* study, which is not necessarily the evidence for this occurring in a *Pf* infection.

Thom Eisele

Reviewer #2:

Remarks to the Author:

In the submitted manuscript, "The density, temporal dynamics and infectiousness of subpatent *P. falciparum* infections", the authors present the results of a literature review to understand important aspects of the detectability of malaria infections by conventional diagnostics—and the relative infectiousness of those appearing as patent or subpatent—across a range of transmission settings. A particular novelty of this work is that the focus is on pooling studies that used quantitative molecular techniques to measure parasite densities (in addition to deploying conventional diagnostics). I am impressed by the scope of the datasets assembled in this review, but find that overall the analysis feels incomplete in ways that for each sub-section (detailed below) seems tied together a common factor: the lack of a clear picture of how parasite densities evolve and fluctuate over the entire duration of an infection in a non-naïve host in the two principal settings (low transmission and moderate-to-high transmission). One reason we lack this picture (besides the cost of qPCR + genetic longitudinal studies with short cadence) is because ethical study designs require prompt treatment of symptomatic infections (or indeed all infections potentially). To make sense of the limited and heterogeneous data we do have requires, in my opinion, either a complex statistical analysis or a complex mechanistic modelling analysis, i.e., something beyond the descriptive statistics presented here.

Section concerning the parasite density distributions of patent and sub-patent infections

- My main concern is that the analysis presented is insufficiently detailed in the exploration of sensitivity of the conventional diagnostics (the hard threshold rather than a soft decay function towards lower densities) and that it ignores specificity. The outliers in each case in Fig 1b beg for an explanation and to be understood in the larger picture (e.g. relation to infectious state): for example, the patent infections at <1 parasite per mL, are these just lucky 'errors' (one might call them "false false positives", i.e., they reflect something about the specificity of the test), or are the microscopists seeing high gametocyte densities (how are these being coded?), or are they just lucky in the sense of finding a lone blood stage parasite in the slide (soft decay function)? Presumably many of these studies have data on mis-classifications (patent but zero qPCR) which could add information here.

- I'm also concerned about the difference between microscopy and RDT as probes of infection / infection history; in the context of patients evolving between patent and sub-patent states over the course of an infection the lagging and persistence of the HRP2 densities (or other) for RDT tests seems to have great consequences for what types of infections each test will pick up as a function of their sensitivity. The comparison between the qPCR densities of RDT and microscopy sub-patents in the SI doesn't convince me that it's fine to lump these together whether for analysis or discussion. For sure there will be many infections for which they agree; my concern would be regarding differences in density (and unobserved stage of infection) where they don't agree. A more illuminating comparison would be something like a KS test on the empirical density distributions for those cases; also, if I read Fig S1 correctly there are only four individuals being compared in each of the China and Haiti studies so any test would be ludicrously underpowered to detect a difference.

Section concerning the analysis of cohort studies

- It's noted that the cohort studies varied in terms of treatment rates; it would seem to me that treatment could potentially have a very large impact of the observations so I would like to see much more detail on this.

- "Since standard PCR for infection multiplicity does not assess the relative density of different parasite clones, no analyses could be performed on clone-specific density fluctuations" – This is one place where the limitations of purely descriptive statistic modelling shows up: a Bayesian model with latent

densities for each genotype would provide a way to start learning what the time course of a single infection in a non-naïve host might look like.

- Another place where further analysis seems warranted is with regard to the plots shown in Fig 4. The contribution of a cross-sectional sub-patent status to the probability of a later patent status is described in terms of two components: probability of developing into a patent infection and increased probability of receiving a later infection owing to a greater inherent exposure. It seems like the curves in Fig 4 are illustrative of the relative size of these two components: the initial gap between curves being the former and the final gap being the latter. Presumably the introduction of a statistical model would allow these to be parameterised and quantified informatively. That said, I find this figure confusing: this is probability of a future positive sample at X-axis weeks, right? In which case how can the sub-patents be non-zero at 0 weeks follow up (as in the top two panels)?

Section concerning the studies of populations in seasonal or declining transmission settings

- "Surveys from multiple time points in areas with seasonal variation in malaria transmission provide an indirect way to look at changing detectability of infections with duration, since during the high transmission season, a larger proportion of infections will have been recently acquired" – It seems like this really depends on what we believe about the time course of infections: if infections have a long tail of low parasite densities this will be true, otherwise not necessarily; if the tails is moderate in length or heterogenous, then ... ?

- "The decrease in average parasite density as an infection progresses is clearly seen in artificially induced untreated malariatherapy infections, but in our analyses of cohort studies, we found no strong evidence that this occurs in these semi-immune individuals living in endemic areas." – Again this depends on how we imagine the time course of infections to progress: if all the variation due to suppression is occurring at parasite densities well above the detection threshold we wouldn't expect to observe a difference here.

Section concerning the onwards infectiousness of patent and sub-patent infections

- "In membrane feeding studies, subpatently infected individuals were found to be approximately a third as infectious to mosquitoes as individuals with patent infection" - Again, I would want to confirm that the patency is with regard to just the blood stage parasites or also the gametocytes; I understand that (perhaps due to clustering) transmission can occur from very low gametocyte densities but one would want to be clear about how the microscopic diagnosis classifies infections with subpatent blood stage and patent gametocyte stage.

Response to Reviewers' comments

The density, temporal dynamics and infectiousness of subpatent *P. falciparum* infections.

Reviewer #1 (Remarks to the Author):

This is a well written paper on a very important issue in malaria epidemiology, namely the temporal dynamics of asexual stage parasite densities during a *P. falciparum* infection, as well as the infectivity to mosquitoes associated with such varying densities over time. It is particularly important that the researchers point out the deficiency in assess subpatent infections from cross-sectional studies, which do not allow the assessment of how parasite densities vary over time during an infection. To address this temporal dynamic of infections, the team identified cohort studies with repeated microscopy-PCR measures from a systematic review of the published literature, which to my knowledge represents the first time this has been done in a comprehensive multi study approach. This research is important to malaria epidemiology. The methods and statistical analyses are valid and form a solid foundation from which the conclusions are drawn.

There are a few areas that should be considered to help improve the paper:

- Are there any data in the cohort studies on fever at each follow-up period so that clinical malaria episodes could be estimated and the association with parasite density explored? I'm assuming they would have needed to be treated if they developed symptoms, but this still might provide valuable insight into the natural history of a malaria parasite infection and symptomology.

Thank you for this suggestion, indeed most cohorts (though not all) recorded clinical episodes and treated these. We have now added this information to Table S2. The definition of clinical episodes in all the cohort studies included being slide positive, and some also applied specific parasite density thresholds. The relationship between parasite densities and clinical malaria has been explored before in the cohort in Papua New Guinea <https://www.ncbi.nlm.nih.gov/pubmed/22665809> and this study also highlighted that clinical episodes are usually associated with new genotypes, indicating recent infection (or superinfection). Similar relationships have been established in other settings (<https://www.ncbi.nlm.nih.gov/pubmed/16903881>). Since our manuscript is already long, we chose to focus on the submicroscopic episodes in the cohorts which have been less explored in the past. We assumed that infections occurring after clinical episodes would mostly be new infections given the treatment in these cohorts.

- In assessing the risk of current submicroscopic infected individuals in the cohorts becoming slide-positive at a later follow-up period, compared to PCR-negative individuals, the time periods being assessed would likely impact this relationship. Can the authors please explain this finding a little further?

Thank you for raising this point, it's a good suggestion to think more about the time period here. This comment is related to the reviewer's later comment "Following from the above point, on the observation that submicroscopic infections were a significant predictor of a later microscopy-detectable infection, it appears most likely that the mode of action is that the submicroscopic infection acting is a marker for higher risk of acquiring a new infection in the future, and thus causing a rebound in parasite densities at that time." We have responded in further detail and carried out simulations to look at this below and in related comments from reviewer 2.

- On page 16 (line 1-2), can the authors please explain how they arrived at a ratio of slide prevalence during the low vs. high season as 0.5 to indicate high seasonality? This seems arbitrary. An alternative approach to this analysis would be to include all studies in a simple logistic regression model for the odds ratio of wet vs. dry season on slide prevalence, while including a covariate for highly season vs not.

We first tested for an effect of season in all sites as reported earlier in the paragraph, then chose this cut-off of 0.5 for a subgroup analysis, as it would indicate a reasonable level of seasonality, considering that slide prevalence is not thought to vary greatly by season (e.g. see Smith et al 1993, *Absence of seasonal variation in malaria parasitaemia in an area of intense seasonal transmission. Acta Trop*). If we understand the alternative test suggested by the reviewer correctly, this would be to test for an interaction between season and the degree of seasonality, incorporating the degree of seasonality as a linear variable rather than selecting any cut-off. We agree this is a good idea. Since we are pooling studies from different settings, using aggregate data, we were using a meta-regression model to allow for different sample sizes and random effects by site. We reran our meta-regression model to test whether the effect of season changed linearly with the degree of seasonality, which similarly showed a positive, non-significant trend. We modified the results as follows.

"When all the seasonal survey results were pooled⁴² the ratio of microscopy sensitivity in the high season relative to the low season was 1.07, 95% CI 0.50-2.31 (i.e. a similar fraction of PCR-positive infections was detected by microscopy in the wet season). We further categorised which locations had high seasonal variation in transmission, defined as the slide-prevalence in the low season being less than half of the slide prevalence in the high season (n=10 locations). In this subgroup of the most seasonal settings, there was a greater increase in the fraction of infections detected by microscopy in the high season compared to the low season, although this difference in sensitivity was not significant (the ratio of microscopy sensitivity in the high season relative to the low season was 1.25, 95% CI 0.66-2.39. **The same conclusion was reached if we tested across all sites for a linear effect of the degree of seasonal variation (Figure) on microscopy sensitivity, rather than using the cut-off to identify the most seasonal settings.**"

- The likely reasons are stated as to why individuals in low transmission areas have lower parasite densities, as compared to their counterparts in higher transmission areas. However, it is likely a combination of having few new infectious bites, highly clonal parasite populations, and potentially elevated acquired immunity due to highly heterogeneous transmission over space.

Yes, we agree and have edited the text as follows:

Several hypotheses have been proposed to explain why parasite densities are lower in low transmission settings, **and the answer is likely to be a complex and interacting combination of the**

following factors. Firstly, this may simply be because individuals receive fewer infectious bites and are therefore on average further along in their course of infection where parasitaemia is expected to be lower^{11, 56}. Additionally, populations in areas that are now low transmission but were higher transmission in the past will still have acquired immunity, therefore may be better able to control parasite density than expected based on the current level of malaria exposure. Individuals in areas that have historically low transmission would not be expected to have much acquired immunity, however may harbour low parasite densities because they reside in small geographic pockets of high transmission where immunity is higher⁵³. Another contributing factor could be that the low genetic diversity of parasite populations in low transmission settings enables individuals to rapidly acquire immunity to those parasite clones^{54, 55}.

- One of the primary weaknesses of this study is that it was not possible to account for new Pf infections among the cohort over time. The authors argue that while parasite densities in the malaria therapy studies showed a clear decrease in parasite densities after the initial peak in asexual stage parasites, their analysis of the cohorts did not support this (lines 1-6 on page 26). However, the malaria therapy data also showed a second peak in asexual stage parasite densities when an infected individual was inoculated with a second new infection, especially with a different parasite strain. This would then suggest that the likely reason for not seeing a stable decrease in parasite densities over time in the cohorts in endemic areas was that repeated reinfection of individuals kept densities at various levels over time.

Thank you for raising this, we agree and have modified the paragraph accordingly:

“The decrease in average parasite density as an infection progresses is clearly seen in artificially induced untreated malariatherapy infections¹¹, but in our analyses of cohort studies, **we could not test for this due to frequent reinfection. However, there was a non-significant trend towards higher sensitivity of microscopy in the rainy season** when infections would be expected to be more recent and potentially more detectable, compared to the dry season.”

We agree with the reviewer that not accounting for reinfections is a key limitation: this is due to the fact that most of the cohorts did not have genotyping information. Even when genotype data are available, current standard PCR methods do not produce genotype-specific density estimates. We hope that the future application of more specific qPCR techniques will be able to resolve some of these questions further.

- Following from the above point, on the observation that submicroscopic infections were a significant predictor of a later microscopy-detectable infection, it appears most likely that the mode of action is that the submicroscopic infection acting as a marker for higher risk of acquiring a new infection in the future, and thus causing a rebound in parasite densities at that time.

We agree it would be very interesting to know whether this is the case. It seems likely that the observation stems both from this factor, plus the additional effect of submicroscopic infections fluctuating to a higher, microscopy-detectable density. Reviewer 2 has raised the point that slide-positive infection soon after the initial submicroscopic sample may tell us more about fluctuations in density of the initial infection, while a higher risk of slide-positive infection after a long follow up may result from heterogeneity in exposure. We address this comment further below using simulations.

The cohort study cited to suggest parasite densities vary over time, even with a low chance of acquiring a new infection, is from a *P. vivax* study, which is not necessarily the evidence for this occurring in a *Pf* infection.

Thank you for raising this. The cohort study we referred to was from Vietnam (Nguyen TN, von Seidlein L, Nguyen TV, et al. The persistence and oscillations of submicroscopic *Plasmodium falciparum* and *Plasmodium vivax* infections over time in Vietnam: an open cohort study. *Lancet Infect Dis* 2018; **18**(5): 565-72.).

This cohort studied both *P. falciparum* and *P. vivax* infections, but we have re-read the paper and it is not entirely clear whether the statistic we originally referred to is for both species or just *P. falciparum*: “A recent cohort study in Vietnam found that 7% of untreated low density infections later became high density⁶⁰.”

In that paper, figure 4 does show 2 examples of *P. falciparum* infections which start at a low density and progress to higher density, therefore we have modified our text as below. We have also added an additional reference published since our submission.

“Recent cohort studies in Vietnam and Mozambique found that **some individuals with** untreated low density *P. falciparum* infections later had high density infections⁶⁰. While the possibility of new infections could not be excluded, **the Vietnam study was in a** low transmission area with low reinfection risk, suggesting the lower density infection later increased in density.”

Thom Eisele

Reviewer #2 (Remarks to the Author):

In the submitted manuscript, “The density, temporal dynamics and infectiousness of subpatent *P. falciparum* infections”, the authors present the results of a literature review to understand important aspects of the detectability of malaria infections by conventional diagnostics—and the relative infectiousness of those appearing as patent or subpatent—across a range of transmission settings. A particular novelty of this work is that the focus is on pooling studies that used quantitative molecular techniques to measure parasite densities (in addition to deploying conventional diagnostics). I am impressed by the scope of the datasets assembled in this review, but find that overall the analysis feels incomplete in ways that for each sub-section (detailed below) seems tied together a common factor: the lack of a clear picture of how parasite densities evolve and fluctuate over the entire duration of an infection in a non-naïve host in the two principal settings (low transmission and moderate-to-high transmission). One reason we lack this picture (besides the cost of qPCR + genetic longitudinal studies with short cadence) is because ethical study designs require prompt treatment of symptomatic infections (or indeed all infections potentially).

We thank the reviewer for these positive comments. We entirely agree about the lack of a clear picture of how parasite densities evolve in immune populations, a point also raised by reviewer 1.

Indeed this is due to the lack of qPCR data on genotype-specific parasite densities, and ethical issues. While we already touched upon this in some parts of the manuscript, we have added the following sentence to our discussion for further emphasis:

“Wider application of quantitative molecular techniques in future is likely to provide further insight into the fluctuations of clone-specific densities in endemic populations.”

While this is true, we also agree with the reviewer that treatment during such research studies is likely to alter the patterns which truly occur in endemic populations outside research settings, where treatment access is often limited. Despite the limitations of the current data, and constraints on future studies, we believe our analyses in the current paper have made progress on questions of practical relevance, such as underlying density distributions, detectability, seasonal patterns and relative infectiousness of submicroscopic populations.

To make sense of the limited and heterogeneous data we do have requires, in my opinion, either a complex statistical analysis or a complex mechanistic modelling analysis, i.e., something beyond the descriptive statistics presented here.

There have previously been complex statistical analyses on one of the datasets used here: the cohort study in Ghana, where a model was used together with genotype information to infer the duration of infection per genotype. (Falk 2006 Comparison of PCR-RFLP and Genescan-based genotyping for analyzing infection dynamics of *Plasmodium falciparum* AJTMH).

Such analyses are possible where genotyping information is present, which was not the case in the large majority of studies. However these previous studies had not examined patterns of submicroscopic versus microscopic infection because of the lack of genotype specific densities. We do not believe that further complex statistical and mechanistic modelling techniques would improve this manuscript because they cannot compensate for the lack of these data. We have conducted an additional simulation exercise in response to a suggestion below to look at slide-positivity in individuals over time, which gave some interesting insights but also found that various possible assumptions could not be distinguished with the available data (see below). The advantage of the descriptive statistics presented here is to show an informative and transparent analysis of the data. We agree that further longitudinal studies of asymptomatic patients with genotyping information combined with modelling would be the ideal way to learn more about the temporal dynamics of infections.

Section concerning the parasite density distributions of patent and sub-patent infections - My main concern is that the analysis presented is insufficiently detailed in the exploration of sensitivity of the conventional diagnostics (the hard threshold rather than a soft decay function towards lower densities) and that it ignores specificity.

Thanks for this comment, and we agree that hard thresholds miss out interesting information about this range of intermediate densities where probability of detection is greater than zero and less than one. We have tried to tackle this question with our new Figure 2 – we fitted a Bayesian logistic regression model with a study-level random effect to each dataset, estimating the probability of detection by microscopy as a function of qPCR parasite density. This shows that there is huge variation in the probability of detection by microscopy for a given qPCR density in different studies, highlighting a potential lack of consistency in PCR and microscopy between studies. We moved the seasonality plot (figure 5 in original manuscript) to the supplementary material to make room for this new plot.

The outliers in each case in Fig 1b beg for an explanation and to be understood in the larger picture (e.g. relation to infectious state): for example, the patent infections at <1 parasite per mL, are these just lucky 'errors' (one might call them "false false positives", i.e., they reflect something about the specificity of the test), or are the microscopists seeing high gametocyte densities (how are these being coded?), or are they just lucky in the sense of finding a lone blood stage parasite in the slide (soft decay function)? Presumably many of these studies have data on mis-classifications (patent but zero qPCR) which could add information here.

Thanks for this comment – it's really interesting to think more about these infections. In several of the studies we have the data to explore this further where study authors have measured gametocytaemia and differentiated between asexual parasites and gametocytes in the microscopy estimates. In the Goncalves, Burkina Faso data, there are four individuals that are patent with very low densities by qPCR – only 1/4 has gametocytes (and with a very low density 7 gam/ul), however all four are positive for asexual parasites by microscopy, and all with densities >50p/ul, indicating a likely error in the PCR. Similarly, in the Goncalves, Kenya data, there are 2 patent outliers – one of them has gametocytes, but only 1 gam/ul, but again both have asexual parasites >50p/ul by microscopy. In Hofmann, Tanzania and Mwingira, Tanzania, none of the patent outliers have gametocytes, but both have asexual parasites by microscopy. Contrastingly, in the Ouedraogo, Burkina Faso data, there are six individuals with patent but low qPCR parasite density infections – all of these individuals have gametocytes by QT-NASBA (16 - 503 gams/ul), and all of them have asexual parasites by microscopy (although densities are not given). Unfortunately, there are not enough of these outliers that also have information on gametocytaemia to make any strong conclusions, but in general it appears that these infections are typically due to an inconsistency between parasite density estimates between qPCR and microscopy, rather than being driven by high gametocytaemia driving the microscopy readings.

We have added a sentence in to the results section to elaborate on this comment:

"Infected individuals that are patent but have very low parasite densities by PCR typically have asexual parasite densities by microscopy in the traditional range of this method (>50 parasites/ μ l), indicating a disagreement in density measures between the two tests."

- I'm also concerned about the difference between microscopy and RDT as probes of infection / infection history; in the context of patients evolving between patent and sub-patent states over the course of an infection the lagging and persistence of the HRP2 densities (or other) for RDT tests seems to have great consequences for what types of infections each test will pick up as a function of their sensitivity. The comparison between the qPCR densities of RDT and microscopy sub-patents in the SI doesn't convince me that it's fine to lump these together whether for analysis or discussion. For sure there will be many infections for which they agree; my concern would be regarding differences in density (and unobserved stage of infection) where they don't agree. A more illuminating comparison would be something like a KS test on the empirical density distributions for those cases; also, if I read Fig S1 correctly there are only four individuals being compared in each of the

China and Haiti studies so any test would be ludicrously underpowered to detect a difference.

Thanks for highlighting this. To explore this point further we assessed the specificity of microscopy and RDT in comparison to PCR, with the expectation that there would be more false-positives (RDT+, PCR-) in the datasets where RDT is the routine diagnostic, due to post-parasite-clearance persistence of hrp2. However, this was not seen in the data – all datasets that used RDT had specificity > 99% apart from one study in Tanzania, where specificity was 67% (Moshia et al.).

One of the main drivers of discordance between microscopy and RDT is recent history of treatment (Wu et al. Nature 2015) – and all the surveys (except one) using RDT were done in a very low transmission setting (Zanzibar) where recent history of treatment will be very low. The exception was the study in Tanzania (Mosha et al.) where specificity was indeed poor. Therefore, due to the good specificity of RDT in the majority (5/6) of studies that used this diagnostic, we feel that including them is valid.

We agree that the China and Haiti studies are underpowered and have removed any statistical tests related to these data. We also agree that a KS test is a more appropriate test for this situation. (We also now use a KS test to explore differences between gametocyte density distributions in Figure 6).

We have amended the description in the supplementary materials as suggested by the reviewer: We compared the qPCR parasite densities of individuals with submicroscopic vs. sub-RDT infections for each of these studies. Data from China and Haiti are just presented for visual purposes due to the small sample sizes. We performed a Kolmogorov-Smirnov two-sample test on the data from Uganda and Tanzania to assess the equality of the submicroscopic and sub-RDT parasite density distributions for each study. The distributions were found to not be significantly different ($p > 0.05$).

Section concerning the analysis of cohort studies

- It's noted that the cohort studies varied in terms of treatment rates; it would seem to me that treatment could potentially have a very large impact of the observations so I would like to see much more detail on this.

We agree this is a good idea and have now added available treatment data to the supplementary material (Table S2). The cohort in Papua New Guinea had a much higher rate of treatment than the other cohorts in Senegal and Ghana (although some of this treatment was for *P. vivax* infection). The type of treatment data available varied: 2 cohorts did not record treatment reliably, although one of these reported the proportion of individuals with detectable chloroquine in urine samples. We have added the following to the main text:

“Rates of treatment were higher in the cohort in Papua New Guinea than in other cohort studies with available treatment data (Table S2), but there was no obvious difference in submicroscopic patterns in this cohort.”

- “Since standard PCR for infection multiplicity does not assess the relative density of different parasite clones, no analyses could be performed on clone-specific density fluctuations” – This is one place where the limitations of purely descriptive statistic modelling shows up: a Bayesian model with latent densities for each genotype would provide a way to start learning what the time course of a single infection in a non-naïve host might look like.

This is an interesting idea, and we can imagine that such techniques could be ultimately powerfully applied to longitudinal qPCR data with genotype-specific densities, allowing for imperfect detectability, missing data etc, once such studies have been undertaken. However we feel it would not be so informative given the less detailed type of data currently available to us, since it would involve adding assumptions about the natural history of infection that we are uncertain about. In the cohort studies the majority do not have data on genotypes, therefore we cannot distinguish new infections. The minority with genotype data do not have clone-specific density data. We would be uncomfortable trying to infer densities with no possibility at present of distinguishing reinfection, nor validating the output. Furthermore even if possible, such an analysis would represent a significant effort in its own right, and we feel it would warrant a separate publication.

- Another place where further analysis seems warranted is with regard to the plots shown in Fig 4. The contribution of a cross-sectional sub-patent status to the probability of a later patent status is described in terms of two components: probability of developing into a patent infection and increased probability of receiving a later infection owing to a greater inherent exposure. It seems like the curves in Fig 4 are illustrative of the relative size of these two components: the initial gap between curves being the former and the final gap being the latter. Presumably the introduction of a statistical model would allow these to be parameterised and quantified informatively.

This is a very interesting suggestion, thank you. To investigate further, we simulated a cohort of individuals with and without variation in exposure to infection, and with and without fluctuations in parasite densities (declining on average over the course of an infection). As in figure 4, we plotted future proportion patent over time by initial infection status (initial time randomly selected once the simulation was at equilibrium) (Figures provided here). This confirmed the reviewer's predictions and some additional points:

- (1) Without variation in exposure (or some other factor) between individuals, one would expect all three lines in the Figure 4 to eventually converge together. The time this takes depends on the frequency of fluctuations below the detection limit and time to reinfection and therefore transmission intensity. In the majority of cohort data presented in our paper, these lines do not converge.
- (2) Heterogeneity in exposure does indeed result in longer term different risk of future patent infection, as predicted by the reviewer. However these longer term differences could also be caused by slow reinfection rates.

Further quantitative conclusions are difficult given the seasonal patterns in the data, and the difficulty in distinguishing the role of rate of reinfection versus fluctuations of the current infection, either of which produce similar patterns.

Figure: simulations of future slide-positivity in 10,000 individuals by initial infection status, average infection rate 5 per person per year. The simulation is run with and without variation in exposure between individuals, and with and without assuming fluctuations in parasite density. Legend as figure 4, main text (red=start slide-positive, blue=start submicroscopic, grey=start negative).

Rather than lengthen the paper with a description of this simulation, we have opted to describe these patterns in the main text using the intuition that the reviewer provided:

“Future slide-positive infection could arise directly from the current submicroscopic infection increasing in density, from variable sensitivity of microscopy or could simply be a marker of an exposed individual who is more likely to contract a future infection. **The short-term changes in slide-positivity after the initial sample are most likely to be due to either fluctuating densities of the current infection or variable microscopy sensitivity, since new infections would take a longer time to accumulate. One study in Papua New Guinea sampled individuals on consecutive days, showing that about 20% of those initially submicroscopic at day 0 became slide positive on the day after (Figure 4). The persistent difference in risk of slide positive infection by initial infection status which occurred in most cohorts throughout follow-up, up to 35 weeks (Figure 4), is less likely to be due to the initial infection (most of which would have cleared by this time point). The cause of this pattern is more likely to be variation in individual exposure, or some other long-term difference between individuals (e.g. immunity suppressing parasite densities).**”

That said, I find this figure confusing: this is probability of a future positive sample at X-axis weeks, right? In which case how can the sub-patents be non-zero at 0 weeks follow up (as in the top two panels)?

Thank you for raising this. These observations which appeared to be at day 0 were actually at day 1, owing to an unusual study design where individuals were sampled on 2 consecutive days every 8 weeks. We agree this was not clear, and have modified the x axis on these panels to start after zero, and added a note in the figure legend: “*The PNG cohort had a first follow up time on day 1 (top 2 panels).*”

Section concerning the studies of populations in seasonal or declining transmission settings - “Surveys from multiple time points in areas with seasonal variation in malaria transmission provide an indirect way to look at changing detectability of infections with duration, since during the high transmission season, a larger proportion of infections will have been recently acquired” – It seems like this really depends on what we believe about the time course of infections: if infections have a long tail of low parasite densities this will be true, otherwise not necessarily; if the tails is moderate in length or heterogenous, then ... ?

It's true, we are assuming that at least some infections are reasonably long, based on evidence (e.g. 210 days - Falk 2006 Comparison of PCR-RFLP and Genescan-based genotyping for analyzing infection dynamics of Plasmodium falciparum AJTMH,). We have added this into the text:

“During the high transmission season, a larger proportion of infections will have been recently acquired, whilst during dry/low transmission seasons, the average age of infections (the time since acquisition **of the most recent parasite clone**) will be older, **assuming that at least some infections have a duration of several months.**”

As long as we assume the season would not alter the natural history of each individual infection, and that there are at least some long tails, then even with heterogeneous duration of infection, on average the most recently acquired infection within individuals must be older in the dry season compared to the wet season.

-“The decrease in average parasite density as an infection progresses is clearly seen in artificially induced untreated malariatherapy infections, but in our analyses of cohort studies, we found no strong evidence that this occurs in these semi-immune individuals living in endemic areas.” – Again this depends on how we imagine the time course of infections to progress: if all the variation due to suppression is occurring at parasite densities well above the detection threshold we wouldn't expect to observe a difference here.

We agree, and have modified the first part of the sentence to clarify. The second part of the sentence is now changed due to a comment by reviewer 1 so that this issue no longer applies. “The decrease in average parasite density as an infection progresses, **so that it is more often below the detection threshold of microscopy**, is clearly seen in artificially induced untreated malariatherapy infections¹¹, but in our analyses of cohort studies, **we could not test for this due to frequent reinfection.**”

Section concerning the onwards infectiousness of patent and sub-patent infections -“In membrane feeding studies, subpatently infected individuals were found to be approximately a third as infectious to mosquitoes as individuals with patent infection” - Again, I would want to confirm that the patency is with regard to just the blood stage parasites or also the gametocytes

Thank you for highlighting this, and we agree. This sentence is in the abstract before we define terminology. We amended this sentence to:

“individuals infected with subpatent asexual parasite densities were found to be approximately a third as infectious to mosquitoes as individuals with patent (asexual parasite) infection”

I understand that (perhaps due to clustering) transmission can occur from very low gametocyte densities but one would want to be clear about how the microscopic diagnosis classifies infections with subpatent blood stage and patent gametocyte stage.

Several of the studies (Goncalves et al. 2016, Ouedraogo et al. 2015, Koepfli et al. 2015) specifically distinguish between asexual stages and gametocytes by microscopy, meaning that the ‘patent/sub-patent’ classification in the analysis is solely based on asexual stages. In other studies, it is less clear how gametocytes were accounted for in the microscopy estimates. However, asexual parasites typically outnumber gametocytes at least 5:1 and by as much as 50:1 [1], meaning we can have some confidence that asexual parasites do indeed exist in a positive slide reading.

[1] Ouedraogo A. L., et al. 2010. The plasticity of *Plasmodium falciparum* gametocytaemia in relation to age in Burkina Faso. *Malar. J.* **9**:281.

Reviewers' Comments:

Reviewer #1:

Remarks to the Author:

After reading the revised paper and responses to the comments from reviewers, I am satisfied with the responses and revisions.

The paper is improved and is now acceptable for publication in my opinion. Nice job.

Reviewer #2:

Remarks to the Author:

The authors are to be commended for such a comprehensive response to my earlier comments, which I'm satisfied they have addressed. I noticed a number of useful insights from the first reviewer as well; overall, I find the new manuscript to be a very worthwhile contribution to the literature.